# The N+ Implementation Details of RLHF with PPO: A Case Study on TL;DR Summarization

**Shengyi Huang**🤗  **Michael Noukhovitch**❖  **Arian Hosseini**❖
**Kashif Rasul**🤗  **Weixun Wang**🌀  **Lewis Tunstall**🤗

🤗Hugging Face
❖Mila, Université de Montréal
🌀Fuxi AI Lab, NetEase
costa.huang@outlook.com

## Abstract

This work is the first to openly reproduce the Reinforcement Learning from Human Feedback (RLHF) *scaling behaviors* reported in OpenAI's seminal TL;DR summarization work (Stiennon et al., 2020). We create an RLHF pipeline from scratch, enumerate over 20 key implementation details, and share key insights during the reproduction. Our RLHF-trained Pythia models demonstrate significant gains in response quality that scale with model size, with our 2.8B, 6.9B models outperforming OpenAI's released 1.3B checkpoint. Our results highlight best practices in data, training, and evaluation for RLHF. We publicly release the trained model checkpoints and code to facilitate further research and accelerate progress in the field at https://github.com/vwxyzjn/summarize_from_feedback_details.

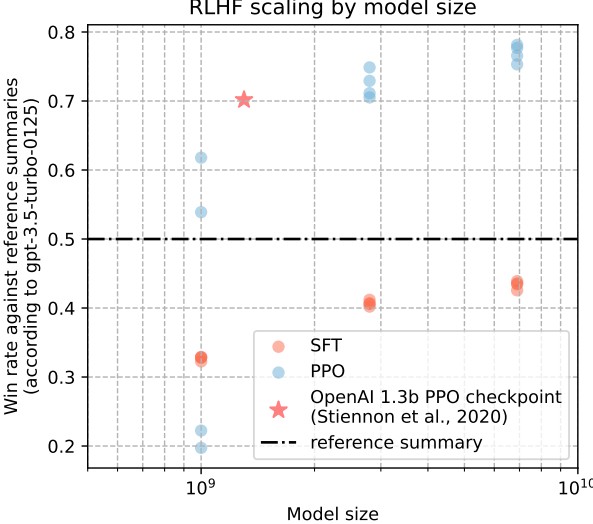

Figure 1: The win rate of our models' summaries over the human-generated reference summaries on the *validation split* of the TL;DR dataset, judged by GPT 3.5. Our SFT / RM / PPO models were trained with four random seeds across the 1B, 2.8B, and 6.9B Pythia (Biderman et al., 2023) model sizes using the same 3e-6 learning rate.

# 1 Introduction

There has been tremendous development in pre-trained large language models (LLMs) over the years (Radford et al., 2018; 2019; Brown et al., 2020; Rae et al., 2021). Given the previous tokens, these LLMs are trained to predict the next token accurately, and they can be prompted to solve a wide range of natural language processing (NLP) tasks. However, the next-token-prediction objective differs from the fundamental chatbot objective of "outputting content that humans prefer". To address this gap, Reinforcement Learning from Human Feedback (RLHF; Stiennon et al., 2020; Ouyang et al., 2022; Bai et al., 2022) was introduced as a pipeline to 1) supervised finetune (SFT) on the domain 2) collect pair-wise human preferences, and train a reward model (RM) to model these preferences then 3) use Reinforcement Learning (RL) to optimize a model against the RM to output content that humans prefer.

It has proven challenging to reproduce OpenAI's RLHF pipeline (Ouyang et al., 2022; OpenAI et al., 2024) in the open-source community for several reasons: 1) RL and RLHF have many subtle implementation details that can significantly impact training stability (Engstrom et al., 2020; Huang et al., 2022; 2024), 2) the models are challenging to evaluate for the instruction following tasks (e.g., evaluating the quality of 800 lines of generated code snippet for a coding task), 3) they take a long time to train and iterate.

This work addresses the aforementioned challenges by taking a step back and reproducing OpenAI's earlier but seminal RLHF work in TL;DR summarization (Stiennon et al., 2020). TL;DR is one of the most popular benchmarks for RLHF methods alongside instruction following tasks such as Anthropic's HH-RLHF (Bai et al., 2022) and AlpacaFarm (Dubois et al., 2023). But summarization tasks are much easier to evaluate than general instruction following tasks because summaries are typically short and bad summaries usually have bad accuracy, coverage, or make-up facts. The reduced context and generation length also mean more efficient training, allowing us to iterate quickly and polish a working RLHF pipeline. Specifically, our contributions are as follows:

**We reproduced the RLHF scaling behaviors in Stiennon et al. (2020).** Our end-to-end pipeline demonstrates that larger models lead to improved ROUGE scores for SFT models, higher validation accuracy for RMs, and higher win rates of the generated summaries over reference summaries for the final RL policies.

**We release a robust, highly reproducible RLHF pipeline** To simplify the setup and improve reproducibility, we use the *same learning rate* for SFT, RM, and RL training, in contrast to the original setup which ran hyperparameter sweeps separately each one. To ensure researchers can reliably reproduce our work, we ran our model training for four random seeds, including failure cases for analysis.

**We enumerate over 20 relevant implementation details and offer detailed insights.** This paper delves into the details of the TL;DR datasets, including their specifications, tokenization processes, and token length distributions. We then detail the training setups, implementation details, and results for both SFT and RM components. Additionally, we explore the details of Proximal Policy Optimization (PPO; Schulman et al., 2017) for RLHF training and how they impact performance. We provide visualizations to compare the behavior of aligned versus base models.

**Our work is fully open source and transparent.** We make our complete source code available at `https://github.com/vwxyzjn/summarize_from_feedback_details`, and release model checkpoints and training metrics in Appendix H.

# 2 Preliminaries

**RLHF** trains a reward model from human preferences and then performs RL training against the reward model (Christiano et al., 2017) for tasks where it is difficult to design a reward function. At a larger scale, RLHF has been used to fine-tune large language models (LLMs) to output contents that align more with human preferences (Ziegler et al., 2019; Stiennon et al., 2020; Ouyang et al., 2022; Bai et al., 2022; OpenAI et al., 2024; Gemini Team et al., 2023).

Table 1: Query pre-processing example. The left example has 512, which is greater than the max query token length of 512, so the pre-processing truncates the last paragraph of the post. Colors show how the contents are tokenized.

| Before: 514 tokens | After: 445 tokens |
|---|---|
| SUBREDDIT: r/relationships
TITLE: Me [19 F] should I be trying to help my brother[16 M] with his life?
POST: This is my first Reddit post and I'm not sure if I'm doing it right

...
I've tried cutting back his computer usage to three hours on weekdays and five hours on weekends but he gets through it. I've tried countless things.
Reddit, should I just leave him be and worry about myself? I love him to death and I miss talking to him like we used to. It's been like this for awhile. Maybe it's puberty, I don't know. I'm at a loss. Someone tell me what to do.
TL;DR: | SUBREDDIT: r/relationships
TITLE: Me [19 F] should I be trying to help my brother[16 M] with his life?
POST: This is my first Reddit post and I'm not sure if I'm doing it right

...
I've tried cutting back his computer usage to three hours on weekdays and five hours on weekends but he gets through it. I've tried countless things.
TL;DR: |

RLHF typically has three steps: 1) train an SFT policy, 2) Collect preference pairs and train an RM, and 3) Train an RL policy with PPO to optimize scores from the RM.

**RL-free approaches** have been proposed to optimize the RLHF objective (Rafailov et al., 2023; Azar et al., 2023; Hong et al., 2024). RLHF with PPO can be quite computationally expensive because the training program typically needs to load 3-4 models into the GPU memory and policy training autoregressively generates online data. To alleviate these problems, researchers have proposed RL-free approaches such as Direct preference optimization (DPO), which implicitly optimizes RLHF by optimizing reward modeling. We describe their specifics and losses in more detail in Appendix A.

**TL;DR Summarization** is the task of summarizing Reddit posts (Völske et al., 2017). Stiennon et al. (2020) showed that optimizing with human preference data can produce summaries that are preferred to those which optimized traditional NLP metrics i.e. ROUGE (Lin & Och, 2004) as well as baseline human-written summaries. The only previous open-source codebase TLRX by Phung et al. (2023) focused mostly on creating an example of an RLHF pipeline and did not reproduce the specific data pipeline, model scores, or scaling behaviour.

## 3 Dataset Details

We start with a solid understanding of the dataset, the tokenization process, and the token lengths. This section provides an in-depth analysis and visualization of the TL;DR datasets from Stiennon et al. (2020), which includes an SFT dataset [1] and a preference dataset [2]. We include more details in Appendix B.

- ➢ **Detail 1: Dataset Specification**. Completions in the preference dataset come from a variety of models and ratings include confidence values that may be taken into account.

---

[1] https://huggingface.co/datasets/vwxyzjn/summarize_from_feedback_tldr_3_filtered
[2] https://huggingface.co/datasets/openai/summarize_from_feedback

Table 2: The number of unique pairs of policies compared differs in each preference dataset split. In particular, notice the validation set contains highly diverse pairs (see Appendix L for details on the exact policy comparisons and their counts).

| Split name | The number of unique pairs of policies compared |
|---|---|
| train | 47 |
| validation | 241 |
| validation_cnndm | 7 |

- ➣ **Detail 2: Do not truncate the sentence, truncate the paragraph**. When the query token length exceeds a preset maximum of 512 tokens, the preprocessing would truncate at the last paragraph instead of a hard truncation limit at 512 tokens. This procedure makes the query coherent. Table 1 shows an example.
- ➣ **Detail 3: Format completions with a leading space, an EOS token at the end, and pad with a special padding token [PAD] instead of just EOS**. Note the prompt ends with TL;DR: which does not include a trailing space, so we need to prepend a leading space in the completion when preparing the query and response (e.g., `long relationship; fell in love with another person; admitted it; would like it to disappear, though it doesn't.<|endoftext|>[PAD][PAD][PAD]`...). It is also important to use a special padding token [PAD] distinct from the EOS token, otherwise EOS can be masked as a padding token and the model won't learn to end summaries with EOS tokens.
- ➣ **Detail 4: SFT and preference datasets have different tokenization length**. Interestingly, the summary lengths in the preference dataset are *not* controlled to be the same. We show visualizations of the tokenization length in Figures 9 and 10 at Appendix B. In particular, The chosen/rejected response token length in the preference dataset can be as long as 169, significantly exceeding the 53 tokens found in the SFT dataset. Also the median chosen response token length is 32, which is slightly longer than that of the rejected response token of 30.
- ➣ **Detail 5: Pre-tokenize the dataset: right pad the concatenation of queries and responses; left pad the queries**. To pre-tokenize the dataset for training, we right pad the concatenation of queries and responses for SFT and RM training; we also left pad the queries for generations during PPO training.
- ➣ **Detail 6: The validation split of the preference dataset has a lot of OOD data.** As illustrated in Table 2 (see Appendix L for details on the exact policy comparisons and their counts), the sampling policies employed in the preference dataset exhibit significant diversity, which is out of the distribution of the sampling policies used in the train split.

## 4 General Details

We note a few general details used across all training, with more in Appendix C.

- ➣ **Detail 7: Disable dropout to ensure PPO's ratio calculation still works**. We disable dropout during all training, similar to the settings in Ziegler et al. (2019); Huang et al. (2024). This is important for PPO, because when dropout is activated, the log probabilities of tokens are not deterministic. This makes calculating the KL penalty unreliable and means the logprob ratios used in PPO are not equal to 1 during the first epoch, which leads to optimization issues. For consistency, we disable dropout for SFT and RM training.
- ➣ **Detail 8: Setup – Tech stack**, We used the transformers (Wolf et al., 2020) library's implementation of the Pythia models in conjunction with deepspeed's ZeRO Stage 2 (Rasley et al., 2020; Rajbhandari et al., 2020) to help fit the models into the GPU memory; for 6.9B PPO training we also offload the reference policy and reward model to CPU. We launch experiments using accelerate (Gugger et al., 2022) with bf16 mixed-precision training and track them with Weights and Biases (Biewald, 2020). We use 8xH100 machines and always upload the trained models to Hugging Face's model hub[3].

---

[3] https://huggingface.co/models

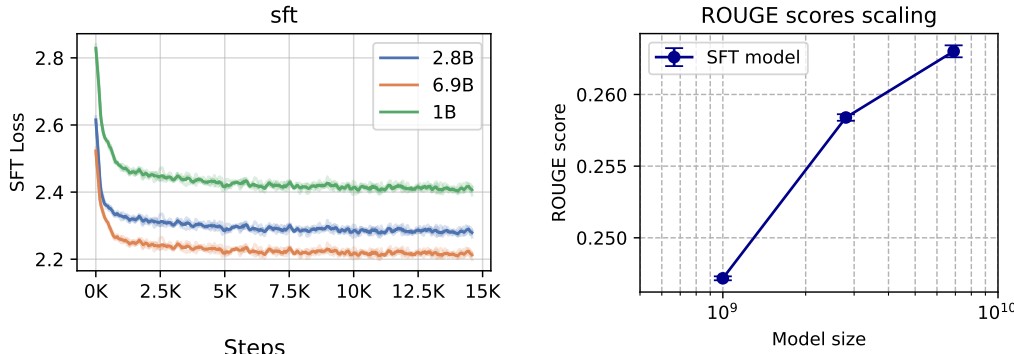

Figure 2: **(Left)** Train loss across model scales for one epoch of the SFT dataset (116k steps) **(Right)** Scaling behaviors of the ROUGE score on SFT validation set

| PPO Response Token Ids | 2325 | 257 | 342 | 33952 | 323 | 608 | · · · | 3553 | 479 | 3815 | 32 | 0 |
|---|---|---|---|---|---|---|---|---|---|---|---|---|
| PPO Response (Decoded) | Be | en | with | GF | for | 5 | · · · | leave | me | alone | ? | <\|endoftext\|> |
| Reward | -4.87 | -4.47 | -5.85 | -4.99 | -7.04 | -6.57 | · · · | -4.63 | -3.93 | -1.83 | 0.24 | 0.65 |

Figure 3: A 1B PPO model's response and its corresponding reward logits from a 1B RM. Here, we use Pythia's tokenizer, so `0` denotes the EOS token and `0.65` is the extracted EOS reward. Notice how the logits of non-EOS tokens are almost always negative – we see this behavior in all the response-reward-logits pairs from all policies and RMs.

## 5 Supervised Finetuning Details

Overall, SFT is fairly straightforward. Our setup closely follows Stiennon et al. (2020), except for a modified learning rate (➤ **Detail 9:**); we show all hyperparameters in Table 3 in Appendix D). We finetune base models on the SFT dataset using standard next-token prediction loss.

As shown in Figure 2, unsurprisingly, larger models have smaller next-token-prediction losses. After finishing the training, we generate summaries from our models and evaluate the ROUGE scores against the reference summaries in the SFT validation set and find a favorable scaling behavior, similar to Figure 14 (a) in Stiennon et al. (2020).

## 6 Reward Modeling Details

Reward modeling, despite its relatively simplicity, has a few important notes to its setup. See Appendix E for more details.

- ➤ **Detail 10:** **Setups**. We follow Stiennon et al. (2020)'s original setting to train the RM, except that we used a different learning rate (see hyperparameters in Table 4 in Appendix E); note the linear head to output reward scalars is initialized with weights according to $\mathcal{N}(0, 1/\sqrt{(d_{\text{model}} + 1)})$.
- ➤ **Detail 12:** **Extract reward from the EOS token** When obtaining the scalar reward, the RM does a forward pass on the sequence and extracts the reward only on the EOS token. This is implemented by finding the first index of the padding token and then minus 1. If the padding token does not exist, the extracted reward will then be logits corresponding to the last token of the sequence – if that token is not the EOS token, its reward won't be used for PPO training, as explained later in PPO's EOS trick at ➤ **Detail 23:**)
- ➤ **Detail 13:** **Most values in the reward logits are non-valid and negative; only the reward logit at the EOS token are valid**. What do the reward logits actually look like in

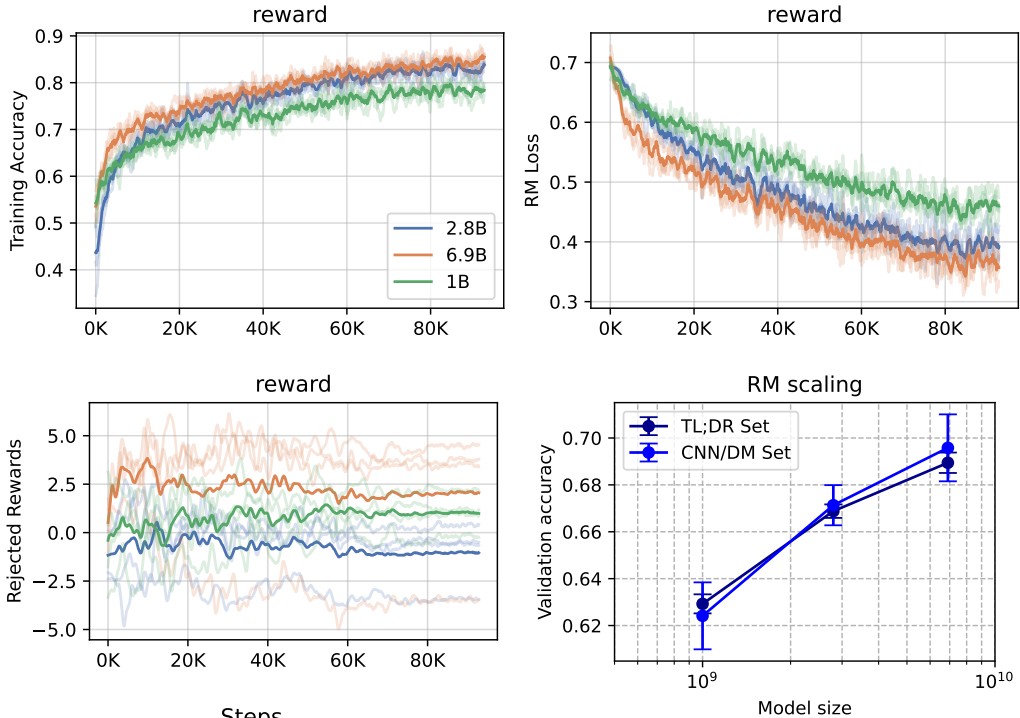

Figure 4: **(Top Left)** The RM training accuracy across the model sizes for one epoch of the train split of the preference dataset (92,858 episodes) **(Top Right)** RM loss **(Bottom Left)** Chosen reward mean values **(Bottom Right)** Scaling behavior of reward modeling on the TL;DR set and CNN/DM validation sets. TL;DR validation accuracy is lower because the validation set has out-of-distribution (OOD) data, explained in ➢ **Detail 6:**.

these trained RMs? We include an example in Figure 3 and notice the logits of non-EOS tokens are almost always negative for all RMs on completions for all policies.

- ➢ **Detail 14: Minor numerical differences between extracting reward with left and right padded queries**. During RM training, the sequences are padded from the right with the shape (B, 638). However, left-padding the query is required for generation in PPO training. The query has shape (B, 512), and after generation (with sequence length = 53), the query and response batch shape becomes (B, 565). As a result, we need to adjust the attention masks during RM forward calls. Left-padding vs. right-padding can introduce minor numerical differences. For instance, in the 6.9B RM, the average reward scalar difference on the SFT dataset between the two padding methods is -0.000544150301720947. This difference is generally negligible.

- ➢ **Detail 15: Reward normalization based on SFT dataset**. Stiennon et al. (2020) suggested that "at the end of training, we normalize the reward model outputs such that the reference summaries from our dataset achieve a mean score of 0." We applied the same procedure by iterating through the SFT dataset and calculating the rewards of the query and reference responses, then calculating the mean reward and setting it as a bias in the reward head.

We show the results of our training (RM training loss, accuracy, and chosen reward value) in Figure 4. The training accuracy and losses appear stable. Overall, larger RMs have higher validation accuracy on both TL;DR and CNN/DM. Note the validation accuracy on the CNN/DM is very encouraging – the RM has never trained on CNN/DM data so this demonstrates good transfer. We notice several additional details in the RM results:

- ➢ **Detail 16: Different batches / confidences have different accuracies**. We calculated the aggregated mean and standard deviation of validation accuracy for each batch, split,

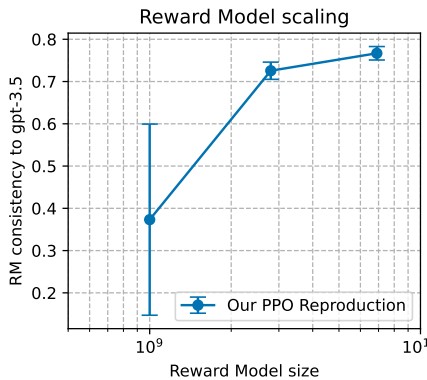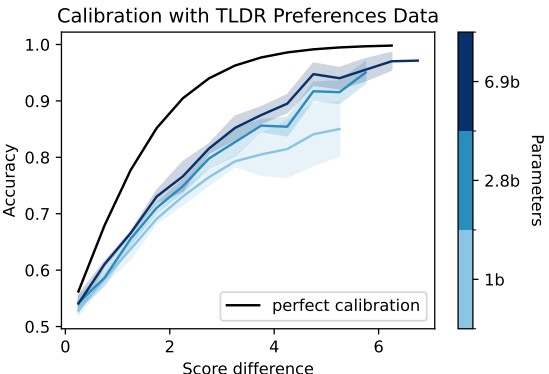

Figure 5: **(Left)** RM agreement rate with GPT3.5 across different model sizes. **(Right)** RM calibration – the black line is the perfect calibration $\frac{1}{1+e^{-\Delta}}$, where $\Delta$ is the the score difference (Equation 2) (Bai et al., 2022).

and confidence in Table 5 at Appendix E. We find different annotated batches have vastly different validation accuracies. Many works choose random subsets of the validation set (Phung et al., 2023; Roit et al., 2023) which can make results incomparable.

- ➤ **Detail 17: Preference consistency with GPT3.5**. To verify whether the RM is overfitting the preference dataset, we use GPT3.5 as an external LLM-judge (Zheng et al., 2023). We compare GPT3.5 and different RM sizes on the model-generated vs human baseline data. As depicted in Figure 5, agreement increases with model size and though it diminishes around 6.9B, ∼ 80% agreement suggests our RM does capture general human preference.
- ➤ **Detail 18: RM calibration**. We follow Bai et al. (2022) to visualize the calibration of the RM on the preference dataset in Figure 5. Overall, we find a positive correlation between accuracy and score difference – models become more accurate as they become more confident (i.e., higher score difference). However, the RMs are still under-calibrated, probably due to the diverse validation set (➤ **Detail 6:**) and different accuracies across these validation sets(➤ **Detail 16:**).
- ➤ **Detail 19: Comparison with DPO as a reward model** We also compare to DPO's RM in Figure 6 and found DPO's validation accuracy to be lower. This may due to several reasons. First, RM training only applies the loss at the EOS token, whereas DPO applies the loss at every completion token. Second, DPO uses a $\beta$ parameter that controls the KL of the language model and therefore also the DPO RM, whereas a regular RM does not. Finally, DPO's objective might be harder to optimize as the model needs to make the logprobs sufficiently different from the base model to change the reward, whereas RM can learn a linear head that can much easier / faster change the value of the reward.

# 7 PPO Details

PPO is well known to be an effective algorithm but can require a variety of specific tricks (Huang et al., 2022). We delve into the main RLHF-specific tricks and explain other, subtle notes in Appendix F.

- ➤ **Detail 20: Setups**. We closely follow Stiennon et al. (2020) in our PPO setup, with our modified learning rate, and detail all hyperparameters in Table 7 in Appendix F.
- ➤ **Detail 21: Train with the SFT dataset and shuffle between epochs**. Stiennon et al. (2020) trains the PPO models for 1M episodes generating from prompts in the SFT dataset. But the `train` split of the SFT dataset is only of size 116,722, so once we've gone through the dataset, we shuffle it and again sample batches without replacement.
- ➤ **Detail 22: Initialize value model from reward model; trained value model looks like a per-token RM**. Following Stiennon et al. (2020), we find it is important to initialize the value network using the reward model. This warm-starting of the value network can greatly improve initial gradients to the policy and reduce drift / alignment tax over training (Noukhovitch et al., 2023). In actor-critic RL training, the value function aims

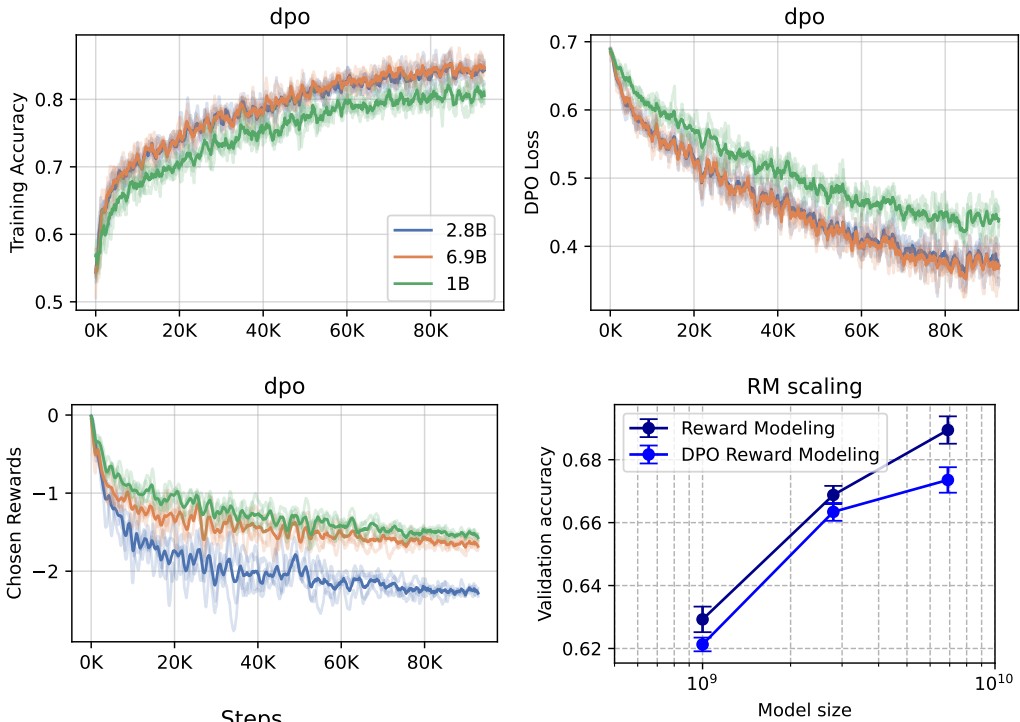

Figure 6: **(Top Left)** DPO RM train accuracy **(Top Right)** DPO train loss **(Bottom Left)** DPO chosen reward mean values **(Bottom Right)** RM and DPO's RM validation set accuracy on TL;DR. We find DPO's accuracy to be lower across all scales (➤ **Detail 19:**).

to predict the end-of-episode return at each timestep / token, effectively acting as a per-token RM. See Figure 11 in Appendix F for the rewards and values of a completion.

- ➤ **Detail 23:"EOS trick" ensures valid rewards from the RM** During training, PPO typically samples a fixed number of tokens but what if the completion does not end with an EOS token? The logits of non-EOS tokens are almost always negative and invalid (➤ **Detail 13:**). If a completion does not end with EOS, the EOS trick sets the reward to a constant -1. This ensures the reward is valid and encourages the models to output shorter responses that end in EOS tokens.

- ➤ **Detail 24:Reward whitening is optional** Huang et al. (2024) note that Ziegler et al. (2019) implement a `whiten` function that normalizes reward values by subtracting the mean followed by dividing by standard deviation. Optionally, `whiten` can shift back the mean of the whitened values with `shift_mean=True`. In each minibatch, PPO could whiten the reward `whiten(rewards, shift_mean=False)` without shifting the mean. In Figure 7, we find reward whitening 1) makes the win rate against reference summaries a bit lower and 2) makes the completion token length a bit shorter. When controlling for summary length, we find it more challenging to compare the results. We therefore consider reward whitening to be optional as it can shorten completions but might not change summary quality.

- ➤ **Detail 25: Advantage whitening is helpful** Similar to practices identified in Engstrom et al. (2020); Andrychowicz et al. (2021); Huang et al. (2022), PPO in Stiennon et al. (2020) whitens the advantages with the shifted mean `whiten(advantages)`.

We train and plot several of PPO's learning curves in Figure 12 in Appendix F. Overall, we find that PPO does well to optimize the RLHF objective, and the average reward goes up. Interestingly, the larger the model, the less the model changes (in KL) to achieve reward. Similar to Gao et al. (2023), we also see that larger RMs tend to better estimate the true reward.

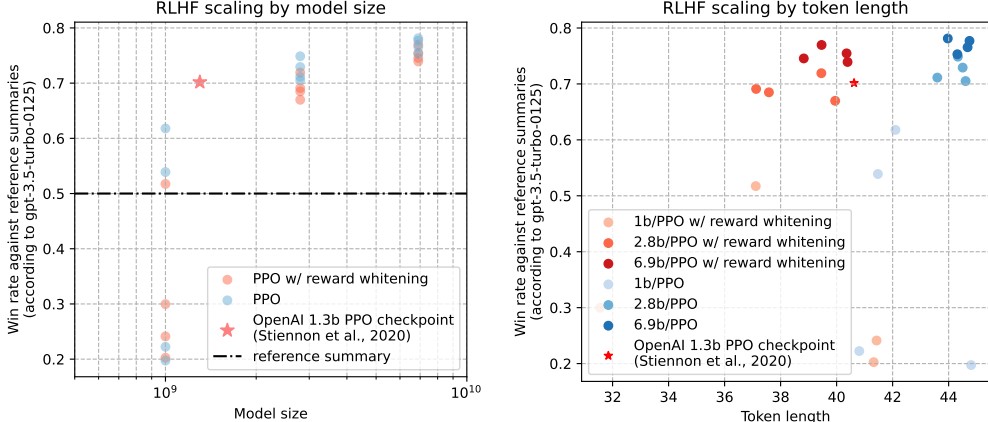

Figure 7: **(Left)** RLHF preference scaling behavior across different model sizes with and without reward whitening ➢ **Detail 24: (Right)** Plots the same data with the x-axis being the average summary token length.

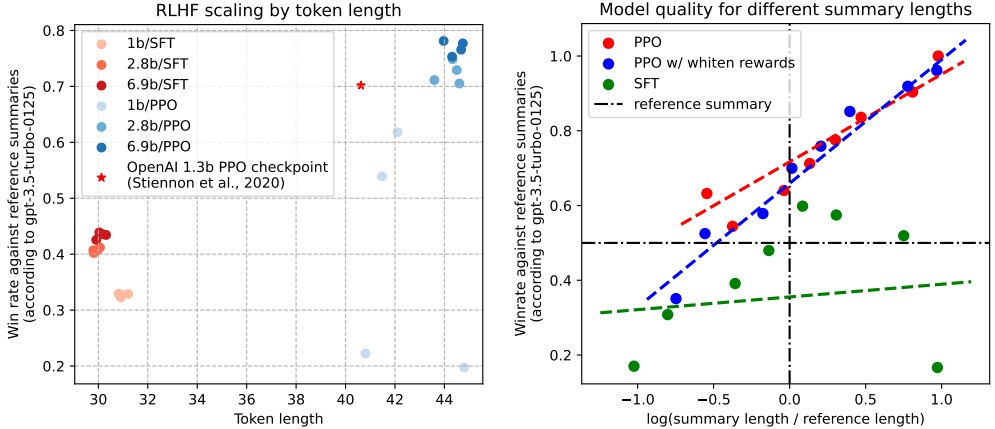

Figure 8: **(Left)** Win rate against reference summaries, plot by average summary token length **(Right)** Win rate of 6.9B SFT and PPO models for different summary lengths (one random seed; see Appendix J for other seeds and model sizes).

We also evaluate the final model checkpoint on the prompts from the validation set of the SFT dataset using GPT 3.5 as a judge against the reference, human-written summaries in Figure 1 (the GPT prompt can be found in Appendix I). We also observed good scaling behaviour for RLHF in terms of win rate. In particular, GPT 3.5 prefers our best 6.9B model to human summaries nearly 80% of the time. To account for correlation between summary length and preference, we plot the win rate against the log(summary length/reference summary length) at Figure 8. Win-rate is correlated with summary length for RLHF models, implying that either true humans preference is correlated with longer summaries or GPT 3.5 is biased in this way. Still, our PPO models almost always outperform SFT across lengths which implies that our models are not simply learning to generate longer summaries but are actually generating higher quality summaries.

## 8 Conclusion

This work presents the first high-fidelity reproduction of OpenAI's RLHF work in TL;DR summarization (Stiennon et al., 2020). We have demonstrated the powerful scaling behavior

of PPO across different Pythia model sizes and shown how RLHF can lead to better summarization models across scales. We offer detailed insights into many implementation specifics and design choices that enabled this successful reproduction, promoting transparency and reproducibility within the research community. We have also noted several interesting results that future work may investigate and elucidate.

Many of our implementation details are not currently standard practices within the RLHF community. We hope this work leads to a collection of best practices in the open-source RLHF community for data, training, and evaluation.

**Author Contributions**

- Shengyi Huang led the overall project.
- Michael Noukhovitch helped discuss and verify early design choices/results, led the analysis of RM calibration plots, and edited the paper.
- Arian Hosseini led the analysis of the length-controlled summary comparisons (e.g., Figure 8), improved visualization in ➤ **Detail 22:**, and edited the paper.
- Kashif Rasul crafted the visualization in Table 8 and edited the paper.
- Weixun Wang plotted the GPT3.5 agreement rate in Figure 5 (left) and Table 5 and edited the paper.
- Lewis Tunstall advised the project.

**Acknowledgments**

The experiments presented in this paper were conducted using NVIDIA H100 nodes in the Hugging Face's compute cluster.

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

## A    RLHF procedure

**Step 1: Train an SFT policy:** The pre-trained LLMs are fine-tuned on the set of human demonstrations using the next-token prediction loss. In this reproduction work, these human demonstrations come from the human summaries of Reddit posts from a filtered TL;DR dataset (Stiennon et al., 2020). In later work, the human demonstrations could come from paid contracted labelers (Ouyang et al., 2022) on a larger variety of tasks.

**Step 2: Collect preference pairs and train an RM:** Various policies, such as the trained SFT policy, are then used to sample completions, and the human labelers would indicate which completions they prefer. Given the preference dataset, we initialize an RM from the SFT policy by adding a randomly initialized linear head that outputs a scalar score. The RM is trained to predict the log probability that a completion would be preferred by the labelers. Specifically, the RM loss is

$$\mathcal{L}_R(r_\phi) = -\mathbb{E}_{(x,y_c,y_r)\sim\mathcal{D}_{\text{PREF}}}\big[\log\sigma(r_\phi(x,y_c) - r_\phi(x,y_r))\big] \tag{1}$$

$$= \mathbb{E}_{(x,y_c,y_r)\sim\mathcal{D}_{\text{PREF}}}\big[\log(1 + e^{r_\phi(x,y_r) - r_\phi(x,y_c)})\big] \tag{2}$$

where $\sigma(x) = \frac{1}{1+e^{-x}}$ is the sigmoid function, $\mathcal{D}_{\text{PREF}}$ the human preference dataset, $x$ the prompt to the model (in this case, the Reddit post), $y_c$ the chosen/preferred completion by a labeler, $y_r$ the rejected completion by the labeler, $\phi$ are the parameters of the RM $r$

**Step 3: Train an RL policy against the RM:** Initializing from the SFT policy, the RL policy then samples completions given prompts and has the RM produce a score based on these completions. The reward of the RL policy then includes this score and a KL penalty to ensure the RL policy does not deviate too much from the SFT policy. Specifically, the reward of the RL problem is

$$R(x,y) = \Big(r_\phi(x,y) - \beta\mathbb{D}_{\text{KL}}\big[\pi_\theta(y\mid x) \mid\mid \pi^{\text{SFT}}(y\mid x)\big]\Big) \tag{3}$$

where $\beta$ is a parameter controlling the strength of the KL penalty, $\theta$ the parameters of RL policy $\pi_\theta$. Then, PPO is used to maximize the RLHF objective $\max_{\pi_\theta}\mathbb{E}_{x\sim\mathcal{D}_{\text{SFT}},y\sim\pi_\theta(y|x)}R(x,y)$, where $\mathcal{D}_{\text{SFT}}$ is the prompts in the SFT dataset.

**RL-free approaches:** The RLHF + PPO pipeline can be quite computationally expensive because 1) the training program typically needs to load 3-4 models into the GPU memory and 2) RL policy training needs online generations and running the RM. To alleviate these two problems, researchers have proposed RL-free approaches (Rafailov et al., 2023; Azar et al., 2023; Hong et al., 2024). One of the most widely-used RL-free approaches is Direct preference optimization (DPO), which has the following loss:

$$\mathcal{L}_{\text{DPO}}(\pi_\theta) = -\mathbb{E}_{(x,y_c,y_r)\sim\mathcal{D}_{\text{PREF}}}\left[\log\sigma\left(\beta\log\frac{\pi_\theta(y_c\mid x)}{\pi^{\text{SFT}}(y_c\mid x)} - \beta\log\frac{\pi_\theta(y_r\mid x)}{\pi^{\text{SFT}}(y_r)}\right)\right]. \tag{4}$$

Note that DPO implicitly does the reward modeling: we can extract the reward score using the following formula:

$$r(x,y) = \beta\log\frac{\pi_\theta(y\mid x)}{\pi^{\text{SFT}}(y\mid x)}. \tag{5}$$

DPO is a more accessible alignment technique that has been implemented in popular RLHF libraries such as TRL von Werra et al. (2020). DPO has also been used to align larger models effectively (e.g., Zephyr 7B (Tunstall et al., 2023), Tulu 70B (Ivison et al., 2023), and Mixtral 8x7B (Jiang et al., 2024)).

## B    Dataset Details – Appendix

### ➤ Detail 1:  Dataset – Specification

The SFT dataset is fairly intuitive – it contains the subreddit, title, post, and reference summary columns. On the other hand, the preference dataset is a lot more nuanced.

The train split of the preference dataset contains the subreddit, title, and post columns; it also contains two sampled summaries, their sampling policies, an internal batch number, the belonging split, which summary the human rater prefers, and optionally, a note or confidence level.

The validation split of the preference dataset contains the same information as above, and *definitely* includes a confidence level. Furthermore, the validation split contains small batches of data for CNN/DM news articles.

➤ **Detail 2: Dataset – Do not truncate the sentence, truncate the paragraph**

The next step is to tokenize the query. The query token goes through the following two transformations (utils/experiment_helpers.py#L196-L199, tasks.py#L98-L165)

1. **Format the query** input string using the following template.
   - SUBREDDIT: r/{subreddit}\n\nTITLE: {title}\n\nPOST: {post}\n\nTL;DR:

2. **Clever truncation** to ensure the query token length is not greater than 512.
   - The formatted query is tokenized using the tokenizer. If the query token length is not greater than 512, it is padded from the left with either padding tokens or repeated white spaces.
   - If the query token length exceeds 512, the pre-processing process will attempt to remove the last paragraph. Specifically, it finds the last index of \n in the post and removes the content after. Table 1 shows an example. This is a much more sophisticated form of truncation compared to a hard truncation on a maximum token length.

3. **No trailing space after** "TL;DR:" to make sure there is no weird generation issues due to the nature of tokenization.

➤ **Detail 3: Dataset – Prepend a leading space to completion; append an EOS token to the completions; use a special padding token [PAD]; do not use EOS token synonymously as [PAD]**

When tokenizing the concatenation of queries and responses for the SFT and preference dataset, we always do the following:

1. Prepends a leading space to the completion, so there is always a space between TL;DR: and the completion such as below.

2. Append an EOS <|endoftext|> token to the completion.

3. When needed to pad the sequence to a maximum length, we always use a special padding token [PAD].

For example, we would add the EOS token and [PAD] token to the reference summary as follows:

```
long relationship; fell in love with another person; admitted it; would like it to
disappear, though it doesn't.<|endoftext|>[PAD][PAD][PAD]...
```

We do *not* recommend using the common practice which uses the EOS token synonymously with the [PAD] token (e.g., tokenizer.pad_token_id = tokenizer.eos_token_id). This is because the EOS token would then be masked out as a padding token during SFT training, and the model would not learn to end a summary – a trained model would often continue to sample summary texts without stopping. This could exacerbate existing issues with RLHF models generating longer outputs (Stiennon et al., 2020; Dubois et al., 2023). With a clear EOS token and padding token distinction, our final trained endpoint always learns to end summaries with the EOS token, as shown in Figure 8.

While Stiennon et al. (2020) choose <|endoftext|> as the EOS token, it may be possible to use another token like <|im_end|>[4] instead as the EOS token. We suspect the key practice is to end the completion with some special token, so the model can learn when to stop.

### ➣ Detail 4: Dataset – SFT and preference datasets have different tokenization length

The SFT dataset had already been filtered such that all the reference summary lengths were controlled – they have a maximum of 48 tokens using the GPT2 tokenizer. In our case, we used Pythia's tokenizer (Biderman et al., 2023), with which the reference summaries have a maximum of 53 tokens. However, an interesting fact is that the summary lengths in the preference dataset are *not* controlled to be the same. Figures 9 and 10 show the length distribution. Several observations:

1. The chosen/rejected response token length in the preference dataset can be as long as 169, significantly exceeding the 53 tokens found in the SFT dataset.
2. The median chosen response token length is 32, which is slightly longer than that of the rejected response token of 30.

### ➣ Detail 5: Dataset – Pre-tokenize the dataset: right pad the concatenation of queries and responses; left pad the queries

To pre-tokenize the dataset for training, we right pad the concatenation of queries and responses and left pad the queries, as shown below.

1. **SFT dataset for SFT training**: we concatenate the query and the reference response together and pad from the right, so during training each sampled batch will have the shape (B, 562).
2. **Preference dataset for RM training**: we concatenate the query-chosen and query-rejected responses together and pad from the right, so during the RM training, each sampled batch will have the shape (B, 638).
3. **Preference dataset for RM evaluation**: During RM evaluation, the sampled batch in the TL;DR splits will have shape (B, 638). Note that in the preference dataset, there is also a split that measures the RM's generalization ability to the CNN/DM dataset, and this split has a much longer token length; in particular, a sampled batch from this dataset will have shape (B, 2021).
4. **SFT dataset for PPO training**: we pad the query from the left to make generations compatible with transformers (since decoder models require left padding for generations), so each sampled batch will have shape (B, 512).

### ➣ Detail 6: Dataset – The validation split of the preference dataset has a lot of OOD data.

As illustrated in Table 2 (see Appendix L for details on the exact policy comparisons and their counts), the sampling policies employed in the preference dataset exhibit significant diversity, which is out of the distribution of the sampling policies used in the train split. As a result, the validation set serves as a great measure of the generalization ability of the (RM).

## C  General Details – Appendix

### ➣ Detail 7: Model – Disable dropout to ensure PPO's ratio calculation still works

We disable the dropout layers during training, similar to the settings in Ziegler et al. (2019); Huang et al. (2024). This is important for PPO training, especially because with dropout activated, the log probabilities of tokens will not be reproducible, making calculating the KL penalty unreliable while also causing the ratios of the PPO to be not 1s during the first

---

[4]https://github.com/openai/openai-python/blob/release-v0.28.0/chatml.md

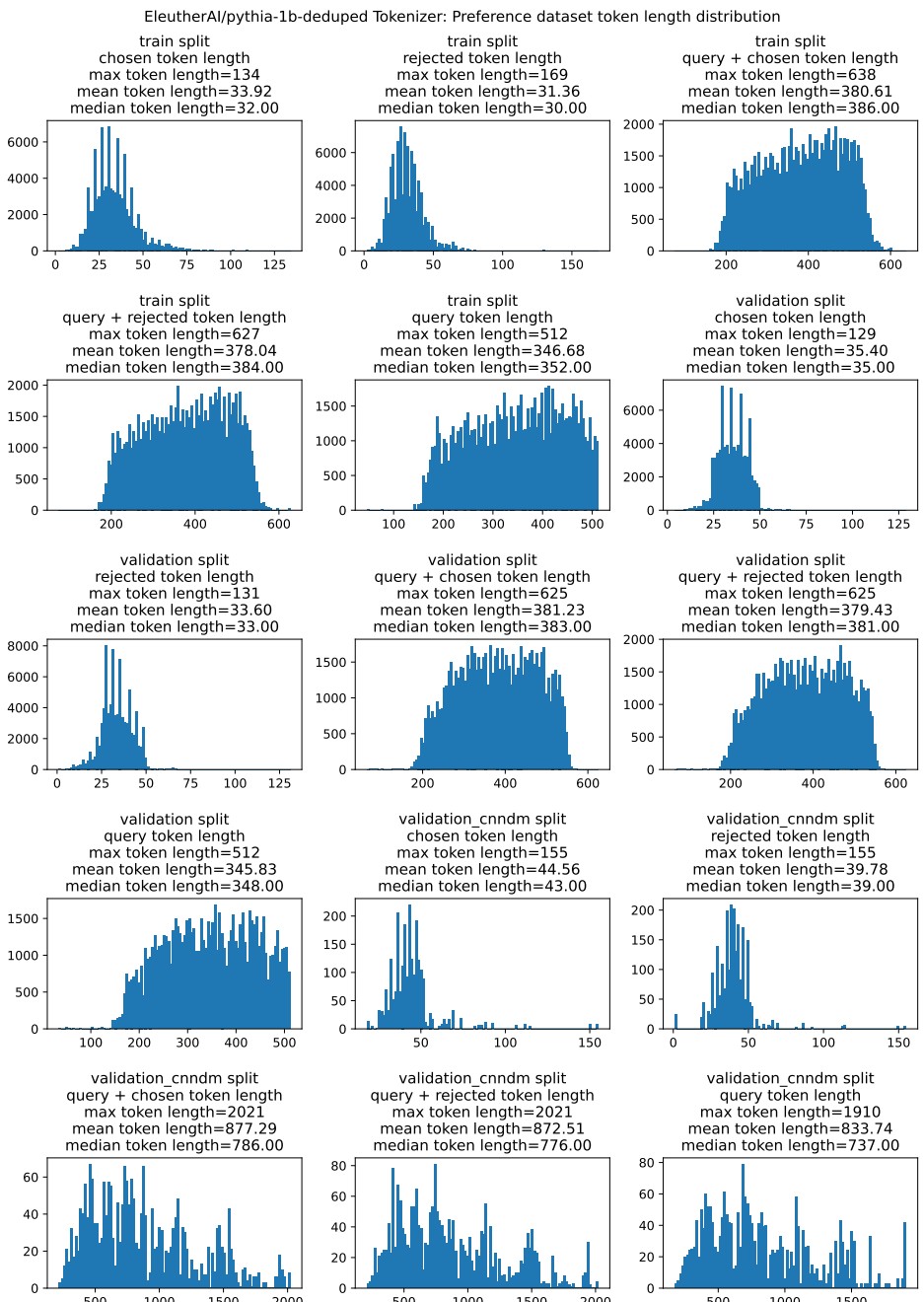

Figure 9: The token length visualization of the preference dataset.

epoch, causing PPO optimization problems. For consistency, we also disable dropout for SFT and RM training.

### ➢ Detail 8: Setup – Tech stack

We used the `transformers` (Wolf et al., 2020) library's implementation of the Pythia models in conjunction with deepspeed's ZeRO Stage 2 (Rasley et al., 2020; Rajbhandari et al., 2020) to help fit the models into the GPU memory; for 6.9B PPO training we also offload the reference policy and reward model to CPU. We launch experiments using `accelerate` (Gugger et al.,

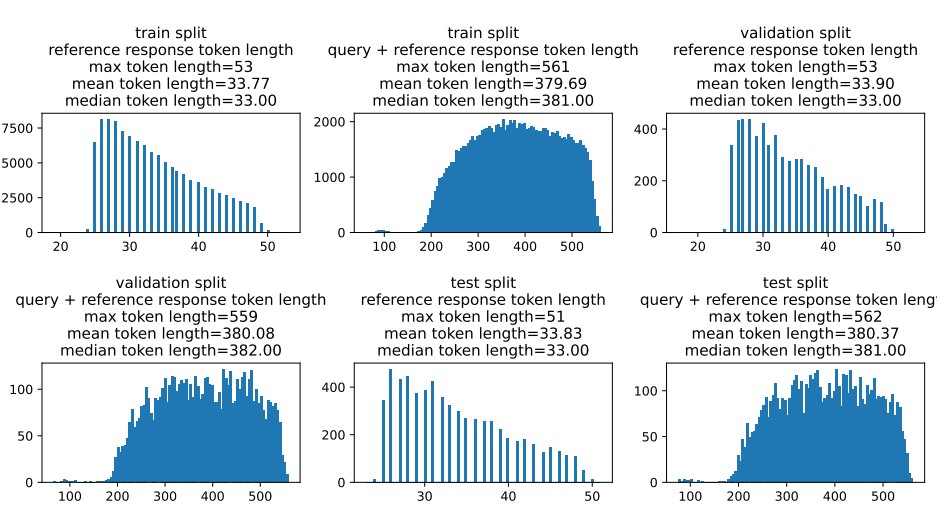

Figure 10: The token length visualization of the preference dataset.

2022) with `bf16` mixed-precision training and track them with Weights and Biases (Biewald, 2020). We use 8xH100 machines and always upload the trained models to Hugging Face's model hub[5].

# D   SFT Details – Appendix

➤ **Detail 9:  SFT Training – Setups**

Our SFT setup closely follows Stiennon et al. (2020), except for a modified learning rate (Table 3).

Table 3: SFT hyperparameters

| Hyperparameter | Default Value |
| --- | --- |
| Number of Train Epochs | 1 (or 116,722 episodes) |
| Optimizer | AdamW ($\epsilon = 1e - 5$, $\mathtt{lr} = 3e - 6$) |
| Scheduler | Cosine |
| Batch Size | 128 |

# E   Reward Model Details – Appendix

➤ **Detail 10:  RM Training – Setups**

We follow Stiennon et al. (2020)'s original setting to train the RM, except that we used a different learning rate (Table 4).

➤ **Detail 11:  RM Training – Reward head initialization**

We follow Stiennon et al. (2020)'s original setting to initialize the RM from the trained SFT model and create a linear heard to output reward scalar with weights initialized according to $\mathcal{N}(0, 1/\sqrt{(d_{\mathrm{model}} + 1)})$ (`query_response_model.py#L106-L108`) [6]

---

[5]https://huggingface.co/models

[6]Note Stiennon et al. (2020) have a minor typo of saying the initialization was according to $\mathcal{N}(0, 1/(d_{\mathrm{model}} + 1))$, but the reference code clearly indicates otherwise.

Table 4: Reward modeling hyperparameters

| Hyperparameter | Default Value |
|---|---|
| Number of Train Epochs | 1 (or 92,858 episodes) |
| Optimizer | AdamW ($\epsilon = 1e-5$, $\mathtt{lr} = 3e-6$) |
| Scheduler | Cosine |
| Batch Size | 64 |

➢ **Detail 12: RM Training – Extract reward from the EOS token**

When obtaining the scalar reward, the RM does a forward pass on the sequence and extracts the reward only on the EOS token. (`reward_model.py`) This is implemented by finding the first index of the padding token and then minus 1. If the padding token does not exist, the extracted reward will then be logits corresponding to the last token of the sequence – if that token is not the EOS token, its reward won't be used for PPO training, as explained later in PPO's EOS trick – ➢ **Detail 23:**).

Note that Stiennon et al. (2020) choose the <|endoftext|> from the base model as the EOS token to extract the reward, but it is possible to use another special token. For example, Andrej Karpathy mentioned that the reward is extracted at <|reward|> in OpenAI's newer GPT systems[7].

➢ **Detail 13: RM Training – Most values in the reward logits are non-valid and negative; only the reward logit at the EOS token are valid**

What do the reward logits actually look like in these trained RMs? We include an example in Figure 3. We noticed the logits of non-EOS tokens are almost always negative in all the response-reward-logits pairs from all policies and RMs.

➢ **Detail 14: RM Training – Minor numerical differences between extracting reward with left and right padded queries**

During RM training, the sequences are padded from the right with the shape (B, 638). However, left-padding the query is required for generation in PPO training. The query has shape (B, 512), and after generation (with sequence length = 53), the query and response batch shape becomes (B, 565). As a result, we need to adjust the attention masks during RM forward calls.

Numerical note: Left-padding vs. right-padding can introduce minor numerical differences. For instance, in the 6.9B RM, the average reward scalar difference on the SFT dataset between the two padding methods is `-0.000544150301720947`. This difference is generally negligible.

➢ **Detail 15: RM Training – Reward normalization based on SFT demonstrations**

Stiennon et al. (2020) suggested that "at the end of training, we normalize the reward model outputs such that the reference summaries from our dataset achieve a mean score of 0." We applied the same procedure by iterating through the SFT dataset and calculating the rewards of the query and reference responses, then calculating the mean reward and setting it as a bias in the reward head.

## E.1 RM training results

The RM training loss, accuracy, and chosen reward value can be found in Figure 4. The training accuracy and losses appear stable. Overall, larger RMs have higher validation accuracy on both TL;DR and CNN/DM sets. Note the validation accuracy on the CNN/DM is very encouraging – the RM has never trained on CNN/DM data! We also performed

---

[7]https://youtu.be/bZQun8Y4L2A?t=956

Table 5: The mean and standard deviation of various metrics of the reward models across four random seeds. The table shows the metric names across different batches, confidences, and splits. There is limited documentation from Stiennon et al. (2020) about these batches and splits, but nevertheless interesting to see this table.

| Metric Names | | | 1B | 2.8B | 6.9B |
|---|---|---|---|---|---|
| **Reward** | | Max | 8.273 ± 0.993 | 5.961 ± 2.45 | 11.75 ± 2.203 |
| | | Mean | 2.114 ± 0.939 | 0.925 ± 2.386 | 4.783 ± 1.545 |
| | | Min | -5.461 ± 1.754 | -5.039 ± 2.547 | -3.016 ± 1.421 |
| | | Std | 1.657 ± 0.086 | 1.361 ± 0.206 | 1.912 ± 0.078 |
| **Validation Accuracy** | | Overall Accuracy | 0.628 ± 0.002 | 0.669 ± 0.003 | 0.689 ± 0.004 |
| | Batch Number | 6 | 0.661 ± 0.016 | 0.682 ± 0.024 | 0.709 ± 0.009 |
| | | 7 | 0.694 ± 0.023 | 0.718 ± 0.011 | 0.732 ± 0.014 |
| | | 8 | 0.598 ± 0.014 | 0.63 ± 0.008 | 0.636 ± 0.009 |
| | | 9 | 0.578 ± 0.005 | 0.687 ± 0.017 | 0.691 ± 0.015 |
| | | 10 | 0.626 ± 0.007 | 0.655 ± 0.015 | 0.69 ± 0.007 |
| | | 11 | 0.508 ± 0.01 | 0.603 ± 0.004 | 0.653 ± 0.021 |
| | | 12 | 0.686 ± 0.007 | 0.697 ± 0.009 | 0.704 ± 0.007 |
| | | 13 | 0.771 ± 0.016 | 0.708 ± 0.013 | 0.745 ± 0.008 |
| | | 14 | 0.577 ± 0.031 | 0.588 ± 0.01 | 0.634 ± 0.011 |
| | | 15 | 0.628 ± 0.021 | 0.699 ± 0.011 | 0.671 ± 0.01 |
| | | 16 | 0.707 ± 0.017 | 0.737 ± 0.002 | 0.761 ± 0.006 |
| | | 17 | 0.752 ± 0.014 | 0.757 ± 0.003 | 0.734 ± 0.018 |
| | | 18 | 0.733 ± 0.015 | 0.741 ± 0.025 | 0.771 ± 0.011 |
| | | 19 | 0.636 ± 0.02 | 0.688 ± 0.012 | 0.714 ± 0.01 |
| | | 20 | 0.671 ± 0.005 | 0.705 ± 0.008 | 0.711 ± 0.007 |
| | | 22 | 0.587 ± 0.006 | 0.632 ± 0.009 | 0.651 ± 0.005 |
| | Confidence | 1 | 0.693 ± 0.012 | 0.758 ± 0.005 | 0.795 ± 0.004 |
| | | 2 | 0.669 ± 0.011 | 0.706 ± 0.012 | 0.718 ± 0.007 |
| | | 3 | 0.635 ± 0.005 | 0.656 ± 0.011 | 0.674 ± 0.003 |
| | | 4 | 0.58 ± 0.005 | 0.562 ± 0.006 | 0.589 ± 0.009 |
| | | 6 | 0.563 ± 0.006 | 0.574 ± 0.012 | 0.581 ± 0.009 |
| | | 7 | 0.568 ± 0.006 | 0.635 ± 0.007 | 0.655 ± 0.008 |
| | | 8 | 0.609 ± 0.011 | 0.691 ± 0.008 | 0.704 ± 0.007 |
| | | 9 | 0.694 ± 0.007 | 0.744 ± 0.005 | 0.765 ± 0.009 |
| | Split Valid | 1 | 0.639 ± 0.003 | 0.667 ± 0.007 | 0.69 ± 0.007 |
| | | 2 | 0.621 ± 0.003 | 0.669 ± 0.003 | 0.688 ± 0.002 |
| **Cnndm Accuracy** | | Overall Accuracy | 0.627 ± 0.013 | 0.665 ± 0.01 | 0.686 ± 0.003 |
| | Batch | Batch0_cnndm | 0.679 ± 0.06 | 0.714 ± 0.027 | 0.723 ± 0.009 |
| | | Cnndm0 | 0.772 ± 0.009 | 0.677 ± 0.017 | 0.714 ± 0.031 |
| | | Cnndm2 | 0.564 ± 0.012 | 0.646 ± 0.013 | 0.666 ± 0.005 |
| | Confidence | 1 | 0.589 ± 0.094 | 0.804 ± 0.043 | 0.815 ± 0.022 |
| | | 2 | 0.641 ± 0.139 | 0.661 ± 0.107 | 0.732 ± 0.036 |
| | | 3 | 0.5 ± 0.037 | 0.771 ± 0.023 | 0.736 ± 0.014 |
| | | 4 | 0.597 ± 0.053 | 0.6 ± 0.028 | 0.615 ± 0.025 |
| | | 6 | 0.671 ± 0.05 | 0.587 ± 0.031 | 0.568 ± 0.02 |
| | | 7 | 0.743 ± 0.095 | 0.646 ± 0.036 | 0.741 ± 0.032 |
| | | 8 | 0.594 ± 0.092 | 0.632 ± 0.056 | 0.662 ± 0.056 |
| | | 9 | 0.65 ± 0.094 | 0.777 ± 0.054 | 0.812 ± 0.061 |
| | Split Valid | 2 | 0.627 ± 0.013 | 0.665 ± 0.01 | 0.686 ± 0.003 |

a comprehensive evaluation of the trained RM on the validation set and calculated the aggregated mean and standard deviation for each batch, split, and confidence in Table 5.

➤ **Detail 16: RM Training – Different batches / confidences have different accuracies**

As shown in Table 5, different annotated batches could have different validation accuracies. Several observations:

1. The 1B model's validation accuracy at batch 11 is 0.508, which is no different from a coin toss

2. The 1B model's validation accuracy at batch 13 is 0.771, a much higher accuracy.

3. The trained RMs generally have high accuracy for high-confidence preference pairs, which makes sense (e.g., the 6.9B model's validation accuracy with accuracy 9 is 0.765).

4. Interestingly, the trained RMs also have high accuracy for very low-confidence preference pairs for some reason (e.g., 6.9B model's validation accuracy with accuracy 1 is 0.795).

**➣ Detail 17: RM Training – Preference consistency rate with GPT3.5**

As per Goodhart's law when a metric becomes the optimization goal, it ceases to be a good metric (Gao et al., 2023). To verify whether RM is overfitting the current dataset's accuracy after training, we introduced GPT3.5 as an external LLM-judge (Zheng et al., 2023). By comparing the preferences of GPT3.5 and RM for the same set of preference data, we assess the actual training effects of RM across different model sizes. As depicted in Figure 5, we have observed the following:

1. For the 1B-sized model, the average preference consistency in multiple random experiments is close to 0.4, indicating that the 1B model has captured a different set of preference, contrary to GPT3.5.

2. The average preference consistency rates for the 2.8B and 6.9B models are 0.726 and 0.767, respectively, both exceeding 0.5. Compared to the 1B model, as the model size increases, RM can exhibit preferences similar to GPT3.5.

3. The difference in average preference consistency rates between the 2.8B and 6.9B models is 0.041, whereas the difference between the 2.8B and 1B models is 0.353. The gains from increasing model size are gradually diminishing (maybe also because the accuracy is already high).

**➣ Detail 18: RM Training – RM calibration**

RMs should predict the log probabilities that humans will prefer one completion versus others; to this end, Bai et al. (2022) propose a visualization technique to see if these probabilities are accurate and well-calibrated. The idea is to plot the score difference between the chosen and rejected pairs in the x-axis and the accuracy of the RM in the y-axis. Intuitively, the larger the score difference, the more confident the model is that one completion is better than the other. We plot the RM calibration in Figure 5.

Overall, we do find a positive correlation between accuracy and score difference – this is a good sign because models become more accurate as they become more confident (i.e., higher score difference). However, the RMs are still under-calibrated, probably due to the diverse validation set (**➣ Detail 6:**) and different accuracies in these validation set (**➣ Detail 16:**).

**➣ Detail 19: RM Training – Comparison with DPO's implicit reward modeling**

We also trained equivalent DPO models to compare the validation accuracy. We use the same hyperparameters used for RM training, except DPO also has a $\beta$ hyperparameter, as shown in Table 6.

During training, we controlled the preference dataset iteration order as well, so this should be a fair comparison of explicit versus DPO's implicit reward modeling losses. The training curves can be found in Figure 6. There are a couple of interesting observations:

Table 6: DPO hyperparameters

| Hyperparameter | Default Value |
|---|---|
| Number of Train Epochs | 1 (or 92,858 episodes) |
| Optimizer | AdamW ($\epsilon = 1e-5$, $\mathtt{lr} = 3e-6$) |
| Scheduler | Cosine |
| Batch Size | 64 |
| $\beta$ (KL Penalty Coefficient for RLHF) | 0.05 |

1. **Validation accuracy regression in DPO**: We found a regression in the validation accuracy in DPO's final evaluation, and this finding holds true across 3 model sizes and 4 random seeds; this suggests DPO's implicit reward modeling may not be equivalent to the regular explicit reward modeling. There are several factors that we suspect may be responsible for this difference. First, regular reward modeling's loss only applies to the EOS token, whereas in DPO, the loss applies to all the tokens. Second, DPO also has the RLHF $\beta$ parameter in the loss, which is not present in regular reward modeling's loss (we chose $\beta = 0.05$ to match PPO's setting). Third, by modeling the reward as the difference in logprobs between model and reference model, DPO's objective may be harder to optimize in practice than the RM objective. Whereas an RM can easily learn large changes in reward using the linear head, DPO must drastically change many tokens' logprobs to do the same.

2. **Decreasing chosen rewards**: DPO's chosen and rejected rewards both generally decrease, whereas regular reward modeling's chosen rewards fluctuate, see Figure 4.

We advocate for more research on how DPO's loss systematically affects RM accuracies.

## F   PPO Details – Appendix

### ➢ Detail 20:  PPO Training – Setups

Our PPO setup closely follows Stiennon et al. (2020), except for a modified learning rate (Table 7).

Table 7: PPO hyperparameters.

| Hyperparameter | Default Value |
|---|---|
| Episodes | 1,000,000 (or ~8.56 epochs) |
| Optimizer | AdamW ($\epsilon = 1e-5$, $\mathtt{lr} = 3e-6$) |
| Scheduler | Linear |
| Batch Size | 512 |
| $\beta$ (KL Penalty Coefficient for RLHF) | 0.05 |
| $\gamma$ (Discount Factor) | 1.0 |
| $\lambda$ (for GAE) | 0.95 |
| $N_{\mathrm{mb}}$ Number of Mini-batches | 1 |
| $K$ (Number of PPO Update Iteration Per Epoch) | 4 |
| $\varepsilon$ (PPO's Policy Clipping Coefficient) | 0.2 |
| $\hat{\varepsilon}$ (Value Clipping Coefficient) | 0.2 |
| $c_1$ (Value Function Coefficient) | 0.1 |
| Value Function Loss Clipping | True |
| Sampling Temperature | 0.7 |

### ➢ Detail 21:  PPO Training – Re-use the SFT dataset and shuffle when reaches the end

Stiennon et al. (2020) trains the PPO models for 1M episodes, but the `train` split of the SFT dataset is only of size 116,722, so an educated guess is that the SFT dataset is re-used

repeatedly during PPO training. Specifically, we should shuffle the SFT dataset and sample from it without replacement; once the dataset is depleted, we should reshuffle it again and sample without replacement; we continue this process until we reach 1M episodes. (`datasets/__init__.py#L27-L39`)

### ➢ Detail 22:  PPO Training – Value model initializes from the reward model; trained value model looks like a per-token RM.

Similar to the settings in Stiennon et al. (2020), we initialize the value network based on the reward model. This warm-starting of the value network can greatly improve initial gradients to the policy and reduce drift / alignment tax over training (Noukhovitch et al., 2023). Because of this, the values generated by the value network will look identical to the example in Figure 3 (➢ **Detail 13:**), where the values of most tokens are negative numbers except for the EOS token.

However, in RL training, the value function would aim to predict the end-of-episode return at each timestep / token, effectively acting as a per-token RM. In Figure 11, we show the rewards and values of a completion, where the 4.5000 is the score from the RM corresponding to the EOS token. The other values in the `rewards` are per-token KL penalty. See `https://wandb.ai/costa-huang/tldr_summarize/runs/9f6t868e/logs` for the full log.

| Value | 2.96 | 2.95 | 3.07 | $\cdots$ | 4.46 |
|---|---|---|---|---|---|
| Reward | -0.02 | -0.02 | 0.00 | $\cdots$ | 4.50 |
| Token | $x_0$ | $x_1$ | $x_2$ | $\cdots$ | EOS |

Figure 11: Reward and values of a completion. The score from the reward model at the EOS token is 4.50 while the rest of `reward` numbers are per-token KL penalty scores.

### ➢ Detail 23:  PPO Training – "EOS trick" to ensure scores from the RM is valid

One interesting phenomenon we observed with the original checkpoint of Stiennon et al. (2020) is that the generated summaries always have less than 48 tokens and also end with an EOS token – this makes the comparison with the reference summaries more fair because the reference summaries are also less than 48 tokens (➢ **Detail 4:**). We suspect the following processes likely achieve it:

1. Always samples a fixed amount of 48 tokens (corresponding to 53 tokens in our reproduction) from the vocabulary (`policy.py#L48`). In particular, the model will continue to sample tokens even if it encounters an EOS token (this means after the EOS token the generations are unconditional).
2. Given the 48 tokens, the script then "truncates" at the EOS token, filling the tokens after the EOS token as padding tokens (`sample.py#L146`, `tasks.py#L57-L62`).
3. This "truncated" response is then passed to the reward model to get a score; if the response does not contain any EOS token, we suspect Stiennon et al. (2020) replaced the score with -1, similar to the procedure described by Ziegler et al. (2019); Huang et al. (2024).

The EOS trick serves a couple of purposes for RL:

1. **Defined reward scores:** It guarantees that the PPO model receives a defined reward score. This is important because the RM only backpropagates loss on the EOS token during training. *Without an EOS token, the completion's reward is undefined.* The EOS trick assigns a constant -1 reward in these cases.
2. **Constraining completion length:** The trick encourages the model to generate concise completions – longer completions that lack an EOS token are penalized with a -1 reward.

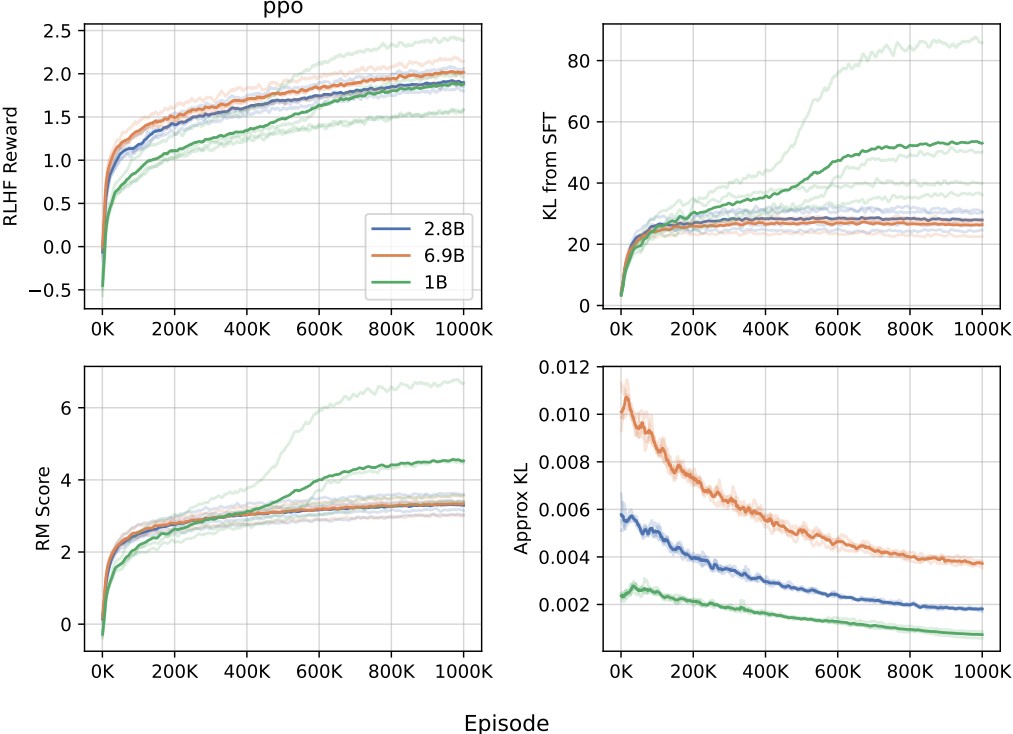

Figure 12: Top left shows PPO's RLHF's reward $R(x, y)$ (Equation 3). The top right figure shows the mean of the sum of per-token KL divergence between the RL and SFT policies. The bottom left shows the scores obtained from the reward model.

Essentially, the EOS trick helps ensure the completion ends with an EOS token, so rewards are well-defined.

➢ **Detail 24: PPO Training – (Optional) Reward whitening**

As indicated in Huang et al. (2024), Ziegler et al. (2019) implement a `whiten` function that looks like below, basically normalizing the values by subtracting its mean followed by dividing by its standard deviation. Optionally, `whiten` can shift back the mean of the whitened values with `shift_mean=True`. In each minibatch, PPO could whiten the reward `whiten(rewards, shift_mean=False)` without shifting the mean (`lm_human_preferences/train_policy.py#L325`).

```python
def whiten(values, shift_mean=True):
    mean, var = torch.mean(values), torch.var(values, unbiased=False)
    whitened = (values - mean) * torch.rsqrt(var + 1e-8)
    if not shift_mean:
        whitened += mean
    return whitened
```

➤ **Detail 25: PPO Training – Advantage whitening**

Similar to practices identified in Engstrom et al. (2020); Andrychowicz et al. (2021); Huang et al. (2022), PPO whitens the advantages `whiten(advantages)` with the shifted mean (`lm_human_preferences/train_policy.py#L338`).

## F.1 PPO training results

We include several PPO's learning curves in Figure 12. We also evaluate the final model checkpoint on the validation set of the SFT dataset using GPT as a judge against the reference summaries in Figure 1 (the GPT prompt can be found in Appendix I). We also conducted an ablation study in which we used reward whitening (➤ **Detail 24:**), and the results are in Figure 7. Finally, to help understand the correlation between summary length and win rate, we plot the win rate against the log(summary length/reference summary length) at Figure 8.

Several observations:

1. **RLHF objective goes up.** Our PPO implementation at least optimizes the RLHF objective, increasing the score total.

2. **Good scaling behaviors.** The preference rate of the PPO models scales nicely with the model checkpoint sizes. In particular, GPT prefers our best 6.9B model nearly 80% of the time.

3. **Over-optimization in 1B models.** For 1B models, the KL divergence seems high (around 50 and 85 for two runs). From an optimization point of view, there is nothing wrong with them because these two runs got higher RLHF Reward $R(x, y)$ (Equation 3), but GPT then judges these two checkpoints to have poor human preference: less than 20% of time GPT prefers them over reference summaries)

   - Upon inspection of these overoptimized samples, we find the PPO policy would concatenate the strings like "Mybestfriendrecentlyblockedmeinsocial-media(atleastonce),anditreallyhurtsme(especiallyafterIwasignoredforaweek). Opinionsandadvicewouldbegreatlyappreciated" (see `https://wandb.ai/costa-huang/tldr_summarize/runs/6qn2rlaq` as an example).

4. **Reward whitening makes the model generate shorter outputs.** We conducted an ablation study with and without reward whitening in Figure 7. Our experiments show that reward whitening makes the model's completions get a lower preference rate, and the completions are shorter than those without reward whitening. However, when inspecting the length-controlled comparisons in Figure 8 (right), the models perform similarly with or without reward whitening in different summary lengths.

5. **PPO models significantly outperform SFT when controlling for length.** As shown in Figure 8 (left), while PPO gets a higher win rate than SFT, the models' responses are generally longer compared to SFT responses, so the summary length is a confounding factor. To address this issue, we control for ratio of summary length to reference length in Figure 8 (right) and show that PPO models outperform SFT models across all summary lengths. We also find that PPO win-rate increases with summary length. This implies that either GPT3.5 prefers longer summaries or longer summaries better optimize true human preference (perhaps implicitly) (Dubois et al., 2023).

Table 8: Sample query, responses from the 1B SFT, PPO, and DPO models; scores are from a 6.9B model. We mark the response tokens ranked top 1 by the pre-trained model blue, meaning unshifted tokens, tokens ranked within the top 3 yellow, meaning marginal tokens, and tokens ranked beyond the top 3 red, meaning shifted tokens (Lin et al., 2024). Essentially, red and yellow tokens highlight what the SFT, PPO, and DPO models would do differently compared to the pre-trained model. We released the source code to load the model and generate this visualization in `https://github.com/vwxyzjn/summarize_from_feedback_details/blob/main/visualize_tokens.py`.

| Type | Content | Score (RM) |
|------|---------|-----------|
| Query | SUBREDDIT: r/AskReddit
TITLE: How do you get someone out of your head?
POST: Hi, I'm 22, and I have been with my girlfriend for 5 years now. We recently moved together. We've always loved each other intensely.
Problem, I recently started to have feelings for another person (a friend). This person has had a boyfriend for now 3 years, and has absolutely no ideas. Those feelings were so strong, it was hard to hide them. After 2 months of me being distant and really sad, my girlfriend forced me to say what was bothering me. I'm not a good liar, and now she knows.
We decided to give us a week alone, I went to my parents.
Now, I'm completely lost. I keep on thinking about this person, and I hate that. I would like for those feelings to go away, to leave me alone. But I can't. What do I do? It's been 3 months now, and I'm just desperate.
TL;DR: | N/A |
| SFT Model Response | I have feelings for a friend, and I'm not sure how to get them out of my head.<|endoftext|> | -3.4151 |
| PPO Model Response | Been with GF for 5 years, recently started to have feelings for another person. I love her deeply however the feelings are driving me crazy. What do I do? Completely lost<|endoftext|> | 2.8743 |
| DPO Model Response | I recently started to have feelings for another person. My girlfriend forced me to say what was bothering me. She now knows. I want those feelings to go away, but I can't. What do I do?<|endoftext|> | 1.354 |
| Reference response | long relationship; fell in love with an other person; admitted it; would like it to disappear, though it doesn't.<|endoftext|> | -1.6587 |
| Base Model Response | How do you get someone out of your head?
A:
I think you're in a situation where you need to get out of your head. You're not in a relationship, and you're not in a relationship with someone who is a good fit for you. You're in a relationship with someone who is not a good fit for you. You're in a | -6.7223 |

## F.2 Visualizing the aligned models vs pre-trained models

Lin et al. (2024) proposed an interesting visualization regarding how aligned models would behave differently from pre-trained models. The idea is to sample a response from the

aligned LLM and check if the pre-trained LLM would greedy sample the same tokens; if so, then color the text blue (unshifted tokens); if the token is within the top 3 probability, color the text yellow; else color the text red (shifted tokens). In simpler terms, the red tokens correspond to what aligned models do differently. We include such visualization of 1B models in Table 8. There are more visualizations of models in the Appendix K. Several observations:

1. **Pre-trained model would continue sampling.** As a result, the generated summary would go significantly beyond the typical lengths of the reference summary or SFT / PPO / DPO summary.

2. **Most tokens are unshifted tokens.** Similar to the findings in Lin et al. (2024), we find most tokens to be unshifted tokens – this means arguably that the summarization ability mostly comes from the pre-trained model.

3. **Fine-tuned models mostly change behaviors at the beginning and the end.** The SFT / PPO / DPO models always alter the initial output and end the summary with an EOS token.

## G   List of model checkpoints and tracked logs

The list of model checkpoints and tracked logs can be found at Table 9.

## H   List of model checkpoints and tracked logs

The list of model checkpoints and tracked logs can be found at Table 9.

## I   GPT as a judge prompt

We modify the GPT as a judge prompt from Rafailov et al. (2023).

```
Which of the following summaries does a better job of summarizing the most \
important points in the given forum post, without including unimportant or \
irrelevant details? Judge based on accuracy, coverage, and coherence.

Post:
<post>

Summary A:
<Summary A>

Summary B:
<Summary B>

FIRST provide a one-sentence comparison of the two summaries, explaining which \
you prefer and why. SECOND, on a new line, state only "A" or "B" to indicate your \
choice. Your response should use the format:
Comparison: <one-sentence comparison and explanation>
Preferred: <"A" or "B">
```

Following Wang et al. (2023); Zheng et al. (2023) we randomize the order of the summaries to remove positional bias in GPT-3.5 Turbo.

## J   Model win rate versus summary lengths

Figure 13 show more plots like Figure 8 (right).

Table 9: List of Hugging Face model checkpoints and tracked Weights and Biases logs.

| Base Model | Type | Seed | 🤗Model Checkpoint | Tracked Wandb Logs |
|---|---|---|---|---|
| EleutherAI/pythia-1b-deduped | ppo | 44413 | 🤗Link | Link |
| | | 55513 | 🤗Link | Link |
| | | 66613 | 🤗Link | Link |
| | | 77713 | 🤗Link | Link |
| | reward | 44413 | 🤗Link | Link |
| | | 55513 | 🤗Link | Link |
| | | 66613 | 🤗Link | Link |
| | | 77713 | 🤗Link | Link |
| | sft | 44413 | 🤗Link | Link |
| | | 55513 | 🤗Link | Link |
| | | 66613 | 🤗Link | Link |
| | | 77713 | 🤗Link | Link |
| EleutherAI/pythia-2.8b-deduped | ppo | 44413 | 🤗Link | Link |
| | | 55513 | 🤗Link | Link |
| | | 66613 | 🤗Link | Link |
| | | 77713 | 🤗Link | Link |
| | reward | 44413 | 🤗Link | Link |
| | | 55513 | 🤗Link | Link |
| | | 66613 | 🤗Link | Link |
| | | 77713 | 🤗Link | Link |
| | sft | 44413 | 🤗Link | Link |
| | | 55513 | 🤗Link | Link |
| | | 66613 | 🤗Link | Link |
| | | 77713 | 🤗Link | Link |
| EleutherAI/pythia-6.9b-deduped | ppo | 44413 | 🤗Link | Link |
| | | 55513 | 🤗Link | Link |
| | | 66613 | 🤗Link | Link |
| | | 77713 | 🤗Link | Link |
| | reward | 44413 | 🤗Link | Link |
| | | 55513 | 🤗Link | Link |
| | | 66613 | 🤗Link | Link |
| | | 77713 | 🤗Link | Link |
| | sft | 44413 | 🤗Link | Link |
| | | 55513 | 🤗Link | Link |
| | | 66613 | 🤗Link | Link |
| | | 77713 | 🤗Link | Link |

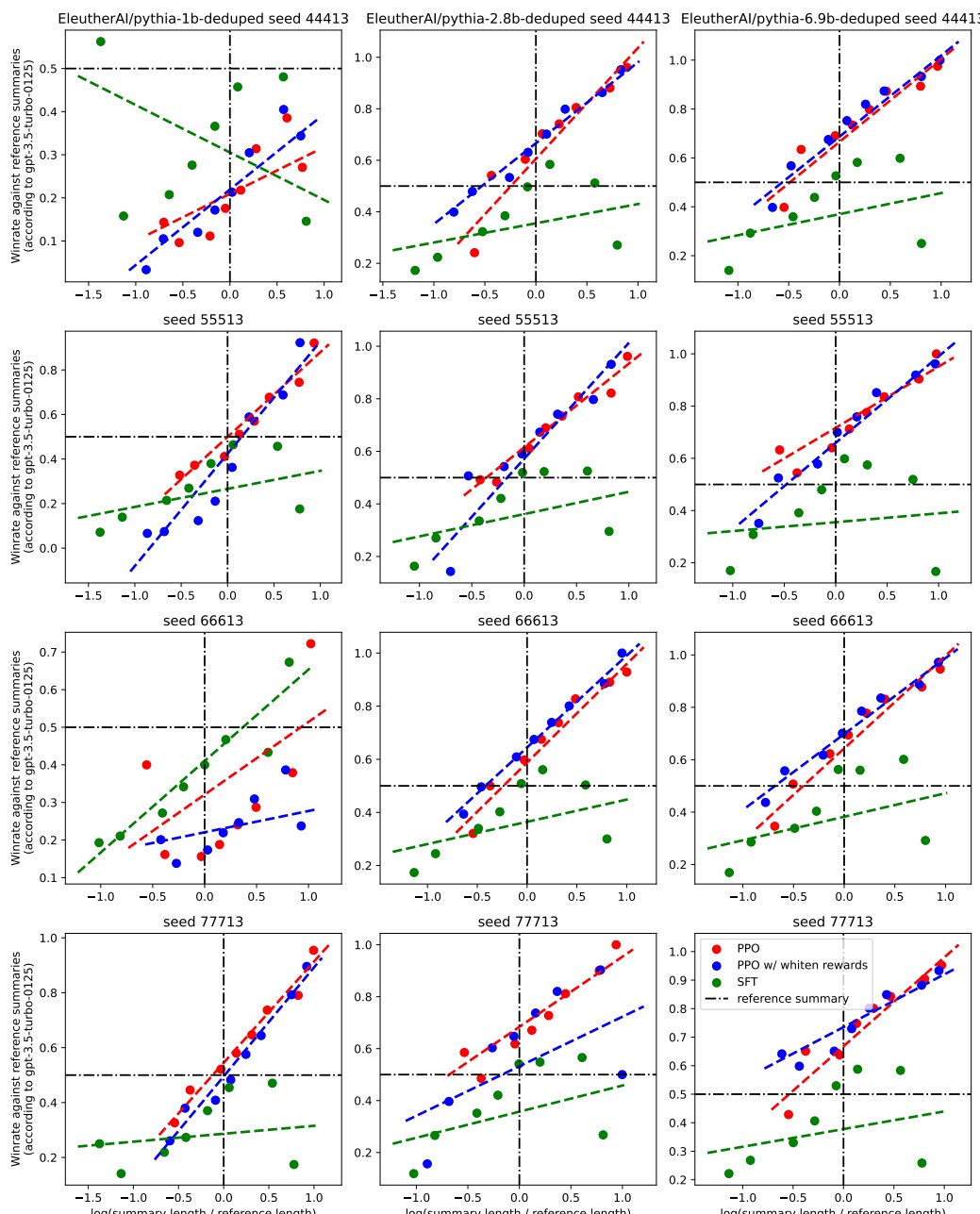

Figure 13: The figure shows the win rate for different summary lengths of the SFT and PPO models across random seeds and model sizes. Note that some of the PPO seeds corresponds to overoptimized policies.

## K   Visualization on aligned models vs base models

This section generates more samples for different-sized models like Table 8.

1. **1B model responses** in Table 10, Table 11, and Table 12
2. **2.8B model responses** in Table 13, Table 14, Table 11, and Table 12
3. **6.9B model responses** in Table 17, Table 18, Table 19, and Table 20

Table 10: Sample query, responses from the 1B SFT, PPO, and DPO models; scores are from a 6.9B model. See Table 8 and Section 8 for semantics on colors.

| Type | Content | Score (RM) |
|---|---|---|
| Query | SUBREDDIT: r/pettyrevenge
TITLE: So, my mom woke me up with a loud TV.
POST: She was in her living room, watching TV. This was at about 8:30 in the morning, and she was exercising. She turned the TV up extra loud to hear it over her excercycle, and woke me up. I went in there asking for her to turn it down. She said she didn't have to; I explained that I always used headphones so she didn't have to deal with my noise and that she should give me a little more respect, given that I paid rent at the time.
She disagreed. I went back to my room, rather pissed off at the lack of equality. I had no lock on my door; but I had a dresser right next to it, so I pulled one of the drawers out enough so that it caused the door to not be openable. Then, I turned my speakers up really loud and blasted Gangnam Style on repeat, with the bass cranked up as high as it could go.
If you hate Gangnam Style for being overplayed, you will see why I chose that particular song. I personally don't mind it. But here's the thing about my bass; it vibrates the walls, making one hell of a lot of noise. Needless to say, my mom was not pleased and shut off the internet. But it was oh so worth it.
TL;DR: | N/A |
| SFT Model Response | Mom woke me up with a loud TV. I turned it down, and she got pissed. I turned it up, and she got pissed.<\|endoftext\|> | -3.9943 |
| PPO Model Response | While watching TV my mom woke me up with a loud racket that vibrated the walls making one hell of a racket, resulting in her shutting off the internet. Very satisfying aftermath<\|endoftext\|> | 1.3313 |
| DPO Model Response | My mom was annoyed at an over-played Gangnam Style song on the morning TV wake-up call, and shut it off. It was worth it.<\|endoftext\|> | -0.6789 |
| Reference response | Mom had the TV on loud and woke me up, didn't care that I'd respected audio levels in the house, so I countered with playing Gangnam Style on repeat with the bass thumping through the walls.<\|endoftext\|> | 1.3359 |
| Base Model Response | I was pissed off at my mom for not giving me more respect, and I was pissed off at her for shutting off the internet.
I'm not sure if I'm the only one who has this problem, but I've been having a problem with my computer for a while now. I've been using it for work, and I've been using it for school. I've been using | -7.4169 |

## L   Details on the comparison pairs in the preference dataset

The comparison pairs and their counts can be found in Table 21, Table 22, Table 23, Table 24, Table 25, Table 26, Table 27.

Table 11: Sample query, responses from the 1B SFT, PPO, and DPO models; scores are from a 6.9B model. See Table 8 and Section 8 for semantics on colors.

| Type | Content | Score (RM) |
|------|---------|------------|
| Query | SUBREDDIT: r/relationships
TITLE: My girlfriend (20f) of two years cheated on me (20m) by kissing two guys at a Halloween party.
POST: Lately her and I have been having a few problems, and these problems have been brought up before a few times. One problem being that I don't show enough affection. I don't tell her she's pretty very often or don't compliment her much. I feel terrible about it, but this time I was really trying to change for her.
For Halloween she went to visit her step brother at a college and I got drunk with my friends and watched movies. Last night (11/1) we got in a huge fight about me not changing and how our relationship won't work out and basically broke up over the phone. So in an effort to try and fix it I drove to her house. She told me how at the parties she went to that two guys kissed her. The first one she pushed away, but the second one I asked her if she kissed him back and she said yes and that she did it because it made her feel wanted, which I guess I haven't been making her feel that way lately. We cried, we talked about everything, we had great sex, and I stayed over at her house just to sleep with her and then snuck out in the morning so her parents wouldn't know.
We both obviously want to work things out but aren't sure if we should. I love this girl, but the more I think about it, all I can think about is her cheating on me, and more importantly, liking it. It makes me sick to my stomach. Should I even try to fix it or would I be better off cutting all ties.
TL;DR: | N/A |
| SFT Model Response | Girlfriend cheated on me by kissing two guys at a Halloween party. I don't know if I should try to fix it or cut all ties.<\|endoftext\|> | 1.6686 |
| PPO Model Response | My girlfriend of two years cheated on me by kissing two guys at a Halloween party. I love her but the more I think about it the more I feel sick about the whole situation. Should I try to fix it or cut all ties?<\|endoftext\|> | 3.7295 |
| DPO Model Response | My girlfriend (20f) cheated on me (20m) by kissing at a Halloween party and we cried and talked about it. Should I try to fix it or should I just cut all ties?<\|endoftext\|> | 2.7493 |
| Reference response | My girlfriend kissed two guys at a Halloween party and liked it. We love each other but don't know what to do about it.<\|endoftext\|> | 0.1145 |
| Base Model Response | I'm not sure if I should try to fix it or cut ties.
I'm not sure if I should try to fix it or cut ties.
I'm not sure if I should try to fix it or cut ties.
I'm not sure if I should try to fix it or cut ties.
I'm not sure if I should try to fix it or | -5.7284 |

Table 12: Sample query, responses from the 1B SFT, PPO, and DPO models; scores are from a 6.9B model. See Table 8 and Section 8 for semantics on colors.

| Type | Content | Score (RM) |
|------|---------|------------|
| Query | SUBREDDIT: r/running
TITLE: One year post-pregnancy, aches and pains in hips and knees, can't even comfortably do day 1 of C25k. Help?
POST: I'll try to keep this short!
**Background**
* I've always been an on again/off again (very casual!) jogger, typically doing 3 - 5 k * My knees have always been finicky, and I went to a physio who thought I had "runner's knee" * Pre-pregnancy, my "runner's knee" would flare up when I got to the 8 - 10 k distance range, even if I had a decent base (doing a C210k type program)
**Current Problem**
I had my baby a year ago, so all in all I haven't run for about 1.5 years. I'm quite slim and have been doing aerobics-style classes for the past year, so I'm not totally out of shape. Body weight exercises, aerobics, bikes and ellipticals are all fine. However, when I run even the tinyiest bit, or even go on a long walk or a hike, my pelvis gets very sore and tight, and my knees start hurting very quickly. I already am doing general squats/lunges/stretching type things.
I'm starting to feel like running just isn't for me anymore. Which is a bummer, because I really enjoy running!
Has anyone had something similar? Can anyone recommend some stretches or exercises that might help? Should I see a Dr? Or should I just see a physio? Not quite sure how to proceed.
Thanks!
TL;DR: | N/A |
| SFT Model Response | I have a very tight pelvis and knees, and I can't run anymore. I'm starting to feel like I'm not running anymore. Should I see a Dr? Or should I just see a physio?<\|endoftext\|> | 0.2434 |
| PPO Model Response | One year post-pregnancy, my knees hurt when I run, can't even comfortably do day 1 of C25k. Has anyone had something similar and what exercises or stretches might help (preferably with a Dr)?<\|endoftext\|> | 4.8225 |
| DPO Model Response | I had my pregnancy year ago and haven't run for about 1.5 years, but when I run my pelvis gets sore and tight, and my knees start hurting very quickly. Should I see a Dr, or should I just see a physio? Not quite sure how to proceed.<\|endoftext\|> | 3.1864 |
| Reference response | I think pregnancy messed with my body, now I can't even run even the smallest amount without pain in my pelvis and knees. I'm fairly certain the problem isn't just that I'm completely out of shape.<\|endoftext\|> | 2.4334 |
| Base Model Response | I have a very tight pelvis, and my knees are very sore. I'm not sure if it's the pregnancy or the running, but I'm not sure if I can run anymore.
A:
I have had the same problem. I have been running for about a year now and I have had a lot of pain in my knees. I have been doing a | -8.9523 |

Table 13: Sample query, responses from the 2.8B SFT, PPO, and DPO models; scores are from a 6.9B model. See Table 8 and Section 8 for semantics on colors.

| Type | Content | Score (RM) |
|------|---------|------------|
| Query | SUBREDDIT: r/AskReddit
TITLE: How do you get someone out of your head?
POST: Hi, I'm 22, and I have been with my girlfriend for 5 years now. We recently moved together. We've always loved each other intensely.
Problem, I recently started to have feelings for another person (a friend). This person has had a boyfriend for now 3 years, and has absolutely no ideas. Those feelings were so strong, it was hard to hide them. After 2 months of me being distant and really sad, my girlfriend forced me to say what was bothering me. I'm not a good liar, and now she knows.
We decided to give us a week alone, I went to my parents.
Now, I'm completely lost. I keep on thinking about this person, and I hate that. I would like for those feelings to go away, to leave me alone. But I can't. What do I do? It's been 3 months now, and I'm just desperate.
TL;DR: | N/A |
| SFT Model Response | I'm in love with a friend, and I can't get her out of my head. I want to get her out of my head, but I can't.<\|endoftext\|> | -3.6299 |
| PPO Model Response | Been with my gf for 5 years, recently developed feelings for another person. Can't stop thinking about him, and hate it. Has anyone been in a similar situation and how did you get past it?<\|endoftext\|> | 3.0524 |
| DPO Model Response | Been with gf for 5 years, moved together, started to have strong feelings for someone else. Made it clear to me that I needed to explain why I was being distant. Now I'm 3 months in and I can't go back to normal. How do I get the strong feeling go away? TL;DR: Been with gf for 5 years, moved together, started to have strong feelings for someone else. Made it clear to me that I needed to explain why | -2.9696 |
| Reference response | long relationship; fell in love with an other person; admitted it; would like it to disappear, though it doesn't.<\|endoftext\|> | -1.6587 |
| Base Model Response | How do you get someone out of your head?
A:
I'm going to assume that you're not in a relationship with this person, and that you're not in a relationship with your girlfriend. If you're in a relationship with your girlfriend, then you're in a relationship with her. If you're not in a relationship with her, then you're not in a relationship with her. If you're in a relationship with your girlfriend, then you're in a | -6.361 |

Table 14: Sample query, responses from the 2.8B SFT, PPO, and DPO models; scores are from a 6.9B model. See Table 8 and Section 8 for semantics on colors.

| Type | Content | Score (RM) |
|---|---|---|
| Query | SUBREDDIT: r/pettyrevenge
TITLE: So, my mom woke me up with a loud TV.
POST: She was in her living room, watching TV. This was at about 8:30 in the morning, and she was exercising. She turned the TV up extra loud to hear it over her excercycle, and woke me up. I went in there asking for her to turn it down. She said she didn't have to; I explained that I always used headphones so she didn't have to deal with my noise and that she should give me a little more respect, given that I paid rent at the time.
She disagreed. I went back to my room, rather pissed off at the lack of equality. I had no lock on my door; but I had a dresser right next to it, so I pulled one of the drawers out enough so that it caused the door to not be openable. Then, I turned my speakers up really loud and blasted Gangnam Style on repeat, with the bass cranked up as high as it could go.
If you hate Gangnam Style for being overplayed, you will see why I chose that particular song. I personally don't mind it. But here's the thing about my bass; it vibrates the walls, making one hell of a lot of noise. Needless to say, my mom was not pleased and shut off the internet. But it was oh so worth it.
TL;DR: | N/A |
| SFT Model Response | My mom woke me up with a loud TV, and I turned it up really loud. She shut off the internet.<\|endoftext\|> | -0.3209 |
| PPO Model Response | My mom woke me up with a loud TV, I turned my speakers up really loud and blasted Gangnam Style. She shut off the internet afterwards.<\|endoftext\|> | 2.4708 |
| DPO Model Response | My loud-volume-waking-up-momma woke me up with the TV turned up loud, so I blasted Gangnam Style on her bass-vibrating walls, to show her who's boss.<\|endoftext\|> | 1.8347 |
| Reference response | Mom had the TV on loud and woke me up, didn't care that I'd respected audio levels in the house, so I countered with playing Gangnam Style on repeat with the bass thumping through the walls.<\|endoftext\|> | 1.3359 |
| Base Model Response | I woke up my mom with a loud TV. She was exercising. I was pissed.
TITLE: I'm not a racist, but I'm not a racist.
POST: I'm not a racist. I'm not a racist. I'm not a racist. I'm not a racist. I'm not a racist. I'm not a racist. I'm not a racist. I'm not a racist. I'm not a racist. I'm not a racist. I'm not a racist. | -5.5731 |

Table 15: Sample query, responses from the 2.8B SFT, PPO, and DPO models; scores are from a 6.9B model. See Table 8 and Section 8 for semantics on colors.

| Type | Content | Score (RM) |
|------|---------|------------|
| Query | SUBREDDIT: r/relationships
TITLE: My girlfriend (20f) of two years cheated on me (20m) by kissing two guys at a Halloween party.
POST: Lately her and I have been having a few problems, and these problems have been brought up before a few times. One problem being that I don't show enough affection. I don't tell her she's pretty very often or don't compliment her much. I feel terrible about it, but this time I was really trying to change for her.
For Halloween she went to visit her step brother at a college and I got drunk with my friends and watched movies. Last night (11/1) we got in a huge fight about me not changing and how our relationship won't work out and basically broke up over the phone. So in an effort to try and fix it I drove to her house. She told me how at the parties she went to that two guys kissed her. The first one she pushed away, but the second one I asked her if she kissed him back and she said yes and that she did it because it made her feel wanted, which I guess I haven't been making her feel that way lately. We cried, we talked about everything, we had great sex, and I stayed over at her house just to sleep with her and then snuck out in the morning so her parents wouldn't know.
We both obviously want to work things out but aren't sure if we should. I love this girl, but the more I think about it, all I can think about is her cheating on me, and more importantly, liking it. It makes me sick to my stomach. Should I even try to fix it or would I be better off cutting all ties.
TL;DR: | N/A |
| SFT Model Response | Girlfriend cheated on me by kissing two guys at a Halloween party. I want to fix it, but I don't know if I should.<\|endoftext\|> | 0.3877 |
| PPO Model Response | My girlfriend of two years cheated on me by kissing two guys at a Halloween party. I love her but the thought of her cheating makes me sick to my stomach. Should I try to fix it or would I be better off cutting all ties?<\|endoftext\|> | 4.1882 |
| DPO Model Response | My girlfriend cheated on me by kissing two guys at a party while she was visiting family. She told me about it during our fight and cried. She says she did it because she felt wanted by the guys, but I think it goes deeper than that. I hate that she did it, and hate myself for not showing her enough affection. Should I try to fix things or cut all ties.<\|endoftext\|> | 3.3642 |
| Reference response | My girlfriend kissed two guys at a Halloween party and liked it. We love each other but don't know what to do about it.<\|endoftext\|> | 0.1145 |
| Base Model Response | I'm in love with a girl, but I'm not sure if I should try to fix things or just cut all ties.
TL;DR: I'm in love with a girl, but I'm not sure if I should try to fix things or just cut all ties.
TL;DR: I'm in love with a girl, but I'm not sure if I should try to fix things or just cut all ties.
TL;DR: I'm in love | -5.5782 |

Table 16: Sample query, responses from the 2.8B SFT, PPO, and DPO models; scores are from a 6.9B model. See Table 8 and Section 8 for semantics on colors.

| Type | Content | Score (RM) |
|---|---|---|
| Query | SUBREDDIT: r/running
TITLE: One year post-pregnancy, aches and pains in hips and knees, can't even comfortably do day 1 of C25k. Help?
POST: I'll try to keep this short!
**Background**
* I've always been an on again/off again (very casual!) jogger, typically doing 3 - 5 k * My knees have always been finicky, and I went to a physio who thought I had "runner's knee" * Pre-pregnancy, my "runner's knee" would flare up when I got to the 8 - 10 k distance range, even if I had a decent base (doing a C210k type program)
**Current Problem**
I had my baby a year ago, so all in all I haven't run for about 1.5 years. I'm quite slim and have been doing aerobics-style classes for the past year, so I'm not totally out of shape. Body weight exercises, aerobics, bikes and ellipticals are all fine. However, when I run even the tinyiest bit, or even go on a long walk or a hike, my pelvis gets very sore and tight, and my knees start hurting very quickly. I already am doing general squats/lunges/stretching type things.
I'm starting to feel like running just isn't for me anymore. Which is a bummer, because I really enjoy running!
Has anyone had something similar? Can anyone recommend some stretches or exercises that might help? Should I see a Dr? Or should I just see a physio? Not quite sure how to proceed.
Thanks!
TL;DR: | N/A |
| SFT Model Response | I'm a runner, but I'm not sure if I'm ever going to be able to run again. Any advice?<\|endoftext\|> | -1.774 |
| PPO Model Response | One year post-pregnancy, aches and pains in hips and knees, can't even comfortably do day 1 of C25k. Has anyone else gone through something similar? What exercises/stretches might help?<\|endoftext\|> | 4.5098 |
| DPO Model Response | One year post-pregnancy, my pelvis and knees hurt when I run, and I'm not exactly out of shape. Has anyone had something similar, and can recommend stretches or exercises? Should I see a Dr?<\|endoftext\|> | 5.5663 |
| Reference response | I think pregnancy messed with my body, now I can't even run even the smallest amount without pain in my pelvis and knees. I'm fairly certain the problem isn't just that I'm completely out of shape.<\|endoftext\|> | 2.4334 |
| Base Model Response | * I'm a runner, but I'm not a runner anymore * I'm not out of shape, but I'm not in great shape either * I'm not sure what to do about my aches and pains
A:
I'm not a doctor, but I have a few suggestions.
I would start with a general stretching routine. I would start with a few minutes of static stretching, then move on to some dynamic stretching. I would | -7.651 |

Table 17: Sample query, responses from the 6.9B SFT, PPO, and DPO models; scores are from a 6.9B model. See Table 8 and Section 8 for semantics on colors.

| Type | Content | Score (RM) |
|---|---|---|
| Query | SUBREDDIT: r/AskReddit
TITLE: How do you get someone out of your head?
POST: Hi, I'm 22, and I have been with my girl-friend for 5 years now. We recently moved together. We've always loved each other intensely.
Problem, I recently started to have feelings for an other person (a friend). This person has had a boyfriend for now 3 years, and has absolutely no ideas. Those feelings were so strong, it was hard to hide them. After 2 months of me being distant and really sad, my girlfriend forced me to say what was bothering me. I'm not a good liar, and now she knows.
We decided to give us a week alone, I went to my parents.
Now, I'm completely lost. I keep on thinking about this person, and I hate that. I would like for those feelings to go away, to leave me alone. But I can't. What do I do? It's been 3 months now, and I'm just desperate.
TL;DR: | N/A |
| SFT Model Response | I have feelings for an other person, and I don't know how to get rid of them.<\|endoftext\|> | -3.2271 |
| PPO Model Response | Been with my girlfriend for 5 years, recently developed feelings for another person. I would like for those feelings to go away, but they won't. I'm lost and don't know what to do<\|endoftext\|> | 3.0259 |
| DPO Model Response | In loving relationship with my girlfriend for 5 years. Started having feelings for another person. My girlfriend found out, and we talked. Now I'm lost, I can't get those other feelings to leave me alone.<\|endoftext\|> | 2.5583 |
| Reference response | long relationship; fell in love with an other person; admitted it; would like it to disappear, though it doesn't.<\|endoftext\|> | -1.6587 |
| Base Model Response | I'm 22, and I have been with my girlfriend for 5 years now. We recently moved together. We've always loved each other intensely. I recently started to have feelings for an other person (a friend). This person has had a boyfriend for now 3 years, and has absolutely no ideas. Those feelings were so strong, it was hard to hide them. After 2 months of me being distant and really sad, my girlfriend forced me to say what was bothering me. I'm not a | -4.9909 |

Table 18: Sample query, responses from the 6.9B SFT, PPO, and DPO models; scores are from a 6.9B model. See Table 8 and Section 8 for semantics on colors.

| Type | Content | Score (RM) |
|------|---------|------------|
| Query | SUBREDDIT: r/pettyrevenge
TITLE: So, my mom woke me up with a loud TV.
POST: She was in her living room, watching TV. This was at about 8:30 in the morning, and she was exercising. She turned the TV up extra loud to hear it over her excercycle, and woke me up. I went in there asking for her to turn it down. She said she didn't have to; I explained that I always used headphones so she didn't have to deal with my noise and that she should give me a little more respect, given that I paid rent at the time.
She disagreed. I went back to my room, rather pissed off at the lack of equality. I had no lock on my door; but I had a dresser right next to it, so I pulled one of the drawers out enough so that it caused the door to not be openable. Then, I turned my speakers up really loud and blasted Gangnam Style on repeat, with the bass cranked up as high as it could go.
If you hate Gangnam Style for being overplayed, you will see why I chose that particular song. I personally don't mind it. But here's the thing about my bass; it vibrates the walls, making one hell of a lot of noise. Needless to say, my mom was not pleased and shut off the internet. But it was oh so worth it.
TL;DR: | N/A |
| SFT Model Response | My mom woke me up with a loud TV. I blasted Gangnam Style on repeat, with the bass cranked up as high as it could go.<\|endoftext\|> | 0.9671 |
| PPO Model Response | My mom woke me up with loud TV. I blasted Gangnam Style loudly through my speakers, making a lot of noise and annoying her. Got some sweet revenge<\|endoftext\|> | 3.9558 |
| DPO Model Response | My mom woke me up with loud TV, because she couldn't hear her excercycle while exercising. She was annoyed with me for having headphones when I paid rent. I blasted Gangnam Style loud enough to vibrate the walls. She turned off the internet.<\|endoftext\|> | 1.3729 |
| Reference response | Mom had the TV on loud and woke me up, didn't care that I'd respected audio levels in the house, so I countered with playing Gangnam Style on repeat with the bass thumping through the walls.<\|endoftext\|> | 1.3359 |
| Base Model Response | My mom woke me up with a loud TV.
SUBREDDIT: r/pettyrevenge
TITLE: I'm a little bit of a dick.
POST: I'm a little bit of a dick.
I'm a little bit of a dick.
I'm a little bit of a dick.
I'm a little bit of a dick.
I'm a little bit of a dick.
I'm | -8.1818 |

Table 19: Sample query, responses from the 6.9B SFT, PPO, and DPO models; scores are from a 6.9B model. See Table 8 and Section 8 for semantics on colors.

| Type | Content | Score (RM) |
|---|---|---|
| Query | SUBREDDIT: r/relationships
TITLE: My girlfriend (20f) of two years cheated on me (20m) by kissing two guys at a Halloween party.
POST: Lately her and I have been having a few problems, and these problems have been brought up before a few times. One problem being that I don't show enough affection. I don't tell her she's pretty very often or don't compliment her much. I feel terrible about it, but this time I was really trying to change for her.
For Halloween she went to visit her step brother at a college and I got drunk with my friends and watched movies. Last night (11/1) we got in a huge fight about me not changing and how our relationship won't work out and basically broke up over the phone. So in an effort to try and fix it I drove to her house. She told me how at the parties she went to that two guys kissed her. The first one she pushed away, but the second one I asked her if she kissed him back and she said yes and that she did it because it made her feel wanted, which I guess I haven't been making her feel that way lately. We cried, we talked about everything, we had great sex, and I stayed over at her house just to sleep with her and then snuck out in the morning so her parents wouldn't know.
We both obviously want to work things out but aren't sure if we should. I love this girl, but the more I think about it, all I can think about is her cheating on me, and more importantly, liking it. It makes me sick to my stomach. Should I even try to fix it or would I be better off cutting all ties.
TL;DR: | N/A |
| SFT Model Response | Girlfriend cheated on me by kissing two guys at a Halloween party. Should I try to fix it or cut all ties?<\|endoftext\|> | 1.3901 |
| PPO Model Response | My girlfriend of two years cheated on me by kissing two guys at a Halloween party. I love her but am conflicted about whether or not to try and fix things. Should I cut ties or try to salvage what we have left<\|endoftext\|> | 5.2579 |
| DPO Model Response | My girlfriend cheated on me by kissing two guys at a Halloween party friendly make out session. She claims she only kissed to feel wanted, but I can't shake the feeling that she cheated and I want to cut all ties.<\|endoftext\|> | 2.0515 |
| Reference response | My girlfriend kissed two guys at a Halloween party and liked it. We love each other but don't know what to do about it.<\|endoftext\|> | 0.1145 |
| Base Model Response | My girlfriend cheated on me with two guys at a Halloween party.
SUB: r/relationships
TITLE: My girlfriend (20f) of two years cheated on me (20m) by kissing two guys at a Halloween party.
POST: Lately her and I have been having a few problems, and these problems have been brought up before a few times. One problem being that I don't show enough affection. I don't tell her she's pretty | -5.0535 |

Table 20: Sample query, responses from the 6.9B SFT, PPO, and DPO models; scores are from a 6.9B model. See Table 8 and Section 8 for semantics on colors.

| Type | Content | Score (RM) |
|---|---|---|
| Query | SUBREDDIT: r/running
TITLE: One year post-pregnancy, aches and pains in hips and knees, can't even comfortably do day 1 of C25k. Help?
POST: I'll try to keep this short!
**Background**
* I've always been an on again/off again (very casual!) jogger, typically doing 3 - 5 k * My knees have always been finicky, and I went to a physio who thought I had "runner's knee" * Pre-pregnancy, my "runner's knee" would flare up when I got to the 8 - 10 k distance range, even if I had a decent base (doing a C210k type program)
**Current Problem**
I had my baby a year ago, so all in all I haven't run for about 1.5 years. I'm quite slim and have been doing aerobics-style classes for the past year, so I'm not totally out of shape. Body weight exercises, aerobics, bikes and ellipticals are all fine. However, when I run even the tinyiest bit, or even go on a long walk or a hike, my pelvis gets very sore and tight, and my knees start hurting very quickly. I already am doing general squats/lunges/stretching type things.
I'm starting to feel like running just isn't for me anymore. Which is a bummer, because I really enjoy running!
Has anyone had something similar? Can anyone recommend some stretches or exercises that might help? Should I see a Dr? Or should I just see a physio? Not quite sure how to proceed.
Thanks!
TL;DR: | N/A |
| SFT Model Response | 1.5 years post-pregnancy, aches and pains in hips and knees, can't even comfortably do day 1 of C25k. Help?<\|endoftext\|> | 1.1442 |
| PPO Model Response | One year post-pregnancy, aches and pains in hips and knees when I try to run even the smallest bit, and I'm not sure if running is for me anymore. Has anyone had something similar? Any advice would be appreciated<\|endoftext\|> | 5.5148 |
| DPO Model Response | One year post-pregnancy, aches and pains in hips and knees when I run, can't even comfortably do day 1 of C25k. Can anyone recommend some stretches or exercises that might help? Should I see a Dr?<\|endoftext\|> | 4.8368 |
| Reference response | I think pregnancy messed with my body, now I can't even run even the smallest amount without pain in my pelvis and knees. I'm fairly certain the problem isn't just that I'm completely out of shape.<\|endoftext\|> | 2.4334 |
| Base Model Response | I'm a runner, but my knees hurt when I run. Any suggestions?
A:
I'm a runner, but my knees hurt when I run. Any suggestions?
I'm a runner, but my knees hurt when I run. Any suggestions?
I'm a runner, but my knees hurt when I run. Any suggestions?
I'm a runner, but my knees hurt when I run. Any suggestions? | -6.3827 |

| policies | count |
|---|---|
| ref, sup2 | 18065 |
| sup2, sup2 | 16273 |
| ref, sup1 | 7947 |
| sup1, sup1 | 7653 |
| sup4_ppo_rm3_kl10, sup4_ppo_rm3_kl10 | 6206 |
| sup4_ppo_rm3_kl20, sup4_ppo_rm3_kl20 | 6098 |
| sup4_6b_t0.7, sup4_6b_t0.7 | 5614 |
| ref, sup3_6b | 1788 |
| ref, sup2_bo8_rm1 | 1786 |
| sup2_bo8_rm1, sup3_6b | 1751 |
| sup2, sup3_6b | 1748 |
| sup2, sup2_bo8_rm1 | 1738 |
| ref, sup4_t0.7 | 1667 |
| sup4_t0.7, sup4_t0.7 | 1330 |
| ref, sup4_ppo_rm3 | 1028 |
| ref, sup3_bo8_rm2 | 958 |
| ref, sup3_ppo_rm1 | 955 |
| sup3_bo8_rm2, sup3_ppo_rm1 | 927 |
| sup4_bo8_rm3, sup4_ppo_rm3 | 775 |
| ref, sup4_bo128_rm3 | 669 |
| sup4_bo128_rm3, sup4_bo256_rm3 | 649 |
| ref, sup3_bo63_rm2 | 480 |
| ref, sup3 | 476 |
| sup3_bo63_rm2, sup3_ppo_rm1 | 470 |
| sup3, sup3_bo8_rm2 | 467 |
| sup3_bo63_rm2, sup3_bo8_rm2 | 464 |
| sup3, sup3_ppo_rm1 | 451 |
| sup4_ppo_rm3, sup4_t0.7 | 441 |
| ref, sup4_bo8_rm3 | 406 |
| sup4_ppo_rm3, sup4_ppo_rm3 | 384 |
| ref, sup4_bo256_rm3 | 340 |
| sup4_bo128_rm3, sup4_bo128_rm3 | 322 |
| sup4_bo64_rm3, sup4_ppo_rm3_kl10 | 255 |
| ref, sup4_ppo_rm3_kl10 | 253 |
| sup4_6b_t0.7, sup4_ppo_rm3_kl20 | 249 |
| sup4_bo128_rm3_6b, sup4_bo256_rm3_6b | 246 |
| ref, sup4_bo128_rm3_6b | 246 |
| ref, sup4_ppo_rm3_kl20 | 245 |
| sup4_6b_t0.7, sup4_ppo_rm3_kl10 | 220 |
| sup4_bo512_rm3, sup4_ppo_rm3_kl20 | 218 |
| ref, sup4_6b_t0.7 | 124 |
| ref, sup4_bo256_rm3_6b | 121 |
| sup4_bo128_rm3_6b, sup4_bo128_rm3_6b | 116 |
| ref, sup4_bo64_rm3 | 70 |
| sup4_6b_t0.7, sup4_bo512_rm3 | 60 |
| sup4_6b_t0.7, sup4_bo64_rm3 | 56 |
| ref, sup4_bo512_rm3 | 53 |

Table 21: The unique comparison pairs and their counts in the *train* split of the preference dataset.

| policies | count |
|---|---|
| ref, sup4_t0.7 | 3252 |
| sup4_t0.7, sup4_t0.7 | 2927 |
| sup4_6b_ppo_rm3_6b_kl15, sup4_6b_ppo_rm4_6b_kl14 | 2669 |
| sup4_ppo_rm3_kl20, sup4_ppo_rm3_kl20 | 2340 |
| sup4_ppo_rm3_kl10, sup4_ppo_rm3_kl10 | 2070 |
| sup4_6b_t0.7, sup4_6b_t0.7 | 1828 |
| pretrain_12b_t.5, ref | 1682 |
| ref, sup4_6b_t0.7 | 1628 |
| ref, sup4_6b | 1167 |
| ref, sup4_6b_ppo_rm4_6b | 1154 |
| ref, sup4_ppo_rm4 | 1141 |
| sup4_12b_t0.7, sup4_ppo_rm4_t.7 | 1097 |
| ref, sup4_ppo_rm3_kl9 | 1084 |
| ref, sup4_12b | 1026 |
| ref, title | 970 |
| ref, sup4_3b | 950 |
| ref, sup4 | 934 |
| pretrain_xl_t.7, ref | 854 |
| pretrain_12b_t.5, sup4_t0.7 | 847 |
| pretrain_12b_t.5, sup4_ppo_rm4_t.7 | 818 |
| pretrain_xl_t.7, sup4_t0.7 | 817 |
| pretrain_12b_t.5, pretrain_xl_t.7 | 816 |
| ref, sup4_ppo_rm4_t.7 | 814 |
| ref, sup4_ppo_rm3_kl6 | 812 |
| ref, sup4_ppo_rm3_kl69 | 810 |
| ref, sup4_ppo_rm3_kl22 | 810 |
| pretrain_12b_t.5, sup4_6b_t0.7 | 799 |
| sup4_6b_t0.7, sup4_ppo_rm4_t.7 | 796 |
| ref, sup4_ppo_rm3_kl2 | 794 |
| pretrain_6b, ref | 786 |
| pretrain_12b, ref | 770 |
| ref, sup4_ppo_rm3_kl260 | 740 |
| ref, sup4_6b_ppo_rm3_6b_kl15 | 726 |
| ref, sup4_12b_t0.7 | 719 |
| sup4_12b_t0.7, sup4_6b_ppo_rm3_6b_kl15 | 715 |
| ref, sup4_3b_t0.7 | 709 |
| pretrain_3b, ref | 707 |
| sup4_6b_ppo_rm4_6b_kl14, sup4_6b_ppo_rm4_6b_kl14 | 670 |
| sup4_6b_ppo_rm3_6b_kl15, sup4_6b_ppo_rm3_6b_kl15 | 669 |
| sup4_3b_t0.7, sup4_t0.7 | 661 |
| pretrain_xl, ref | 567 |
| ref, sup3_6b | 545 |
| ref, sup2 | 543 |
| ref, sup2_bo8_rm1 | 542 |
| sup2_bo8_rm1, sup3_6b | 535 |
| ref, sup4_ppo_rm3_kl10 | 527 |
| ref, sup4_xl_bo64_rouge | 525 |
| sup2, sup2_bo8_rm1 | 523 |
| sup2, sup3_6b | 517 |
| ref, sup4_bo64_rm3 | 507 |

Table 22: The unique comparison pairs and their counts in the *validation* split of the preference dataset. (Part 1)

| policies | count |
|---|---|
| ref, sup4_xl_bo512_rm4_6b | 497 |
| sup4_xl_bo128_rouge, sup4_xl_bo64_rouge | 495 |
| sup4_xl_bo1024_rm4_6b, sup4_xl_bo512_rm4_6b | 483 |
| ref, sup4_bo512_rm3 | 481 |
| ref, sup4_xl_bo64_rm4 | 467 |
| ref, sup4_bo128_rm3 | 464 |
| sup4_xl_bo128_rm4, sup4_xl_bo64_rm4 | 453 |
| pretrain_6b_t.7, ref | 443 |
| ref, sup4_6b_ppo_rm4_6b_kl14 | 442 |
| sup4_6b_ppo_rm4_6b_kl14, sup4_6b_t0.7 | 440 |
| ref, sup4_xl_bo512_rm4 | 439 |
| pretrain_6b_t.7, sup4_6b_ppo_rm4_6b_kl14 | 439 |
| ref, sup4_xl_bo512_rouge | 438 |
| pretrain_6b_t.7, sup4_6b_t0.7 | 436 |
| ref, sup4_xl_bo64_rm4_6b | 436 |
| sup4_xl_bo1024_rm4, sup4_xl_bo512_rm4 | 432 |
| sup4_xl_bo128_rm4_6b, sup4_xl_bo64_rm4_6b | 427 |
| sup4_bo128_rm3, sup4_bo64_rm3 | 417 |
| sup4_xl_bo1024_rouge, sup4_xl_bo512_rouge | 407 |
| sup4_bo512_rm3, sup4_xl_bo1024_rm3 | 403 |
| sup4_ppo_rm3_kl6, sup4_ppo_rm3_kl9 | 387 |
| sup4_ppo_rm3_kl69, sup4_ppo_rm3_kl9 | 380 |
| sup4_ppo_rm3_kl2, sup4_ppo_rm3_kl9 | 351 |
| ref, sup4_ppo_rm3 | 345 |
| sup4_ppo_rm3_kl22, sup4_ppo_rm3_kl9 | 325 |
| sup4_ppo_rm3_kl9, sup4_t0.7 | 322 |
| ref, sup3_ppo_rm1 | 315 |
| sup4_ppo_rm3_kl260, sup4_ppo_rm3_kl9 | 307 |
| ref, sup3_bo8_rm2 | 306 |
| sup3_bo8_rm2, sup3_ppo_rm1 | 302 |
| sup4_bo64_rm3, sup4_ppo_rm3_kl10 | 302 |
| sup4_6b_t0.7, sup4_ppo_rm3_kl10 | 274 |
| ref, sup4_xl_bo128_rouge | 262 |
| sup4_bo8_rm3, sup4_ppo_rm3 | 261 |
| sup4_ppo_rm3_kl6, sup4_ppo_rm3_kl69 | 260 |
| ref, sup4_bo8_rm3 | 248 |
| ref, sup4_6b_p.95 | 248 |
| ref, sup4_xl_bo1024_rm4_6b | 247 |
| sup4_6b, sup4_6b_t0.7 | 245 |
| sup4_xl_bo64_rouge, sup4_xl_bo64_rouge | 244 |
| sup4_6b_p.95, sup4_6b_t0.7 | 244 |
| sup4_6b, sup4_6b_p.95 | 244 |
| sup4_bo128_rm3, sup4_bo256_rm3 | 244 |
| sup4_xl_bo512_rm4_6b, sup4_xl_bo512_rm4_6b | 242 |
| sup4_ppo_rm3_kl22, sup4_ppo_rm3_kl69 | 242 |
| ref, sup4_ppo_rm3_kl20 | 238 |
| sup4_6b_t0.7, sup4_ppo_rm3_kl20 | 234 |
| sup4_ppo_rm3_kl2, sup4_ppo_rm3_kl22 | 234 |
| ref, sup4_xl_bo128_rm4 | 231 |
| sup4_ppo_rm3_kl2, sup4_t0.7 | 228 |

Table 23: The unique comparison pairs and their counts in the *validation* split of the preference dataset. (Part 2)

| policies | count |
|---|---|
| sup4_ppo_rm3_kl22, sup4_ppo_rm3_kl6 | 228 |
| sup4_ppo_rm3_kl2, sup4_ppo_rm3_kl69 | 226 |
| sup4_xl_bo64_rm4, sup4_xl_bo64_rm4 | 224 |
| sup4_bo512_rm3, sup4_ppo_rm3_kl20 | 224 |
| ref, sup4_xl_bo1024_rouge | 219 |
| sup4_ppo_rm3_kl260, sup4_ppo_rm3_kl6 | 219 |
| ref, sup4_xl_bo128_rm4_6b | 218 |
| ref, sup4_xl_bo1024_rm4 | 218 |
| sup4_xl_bo64_rm4_6b, sup4_xl_bo64_rm4_6b | 217 |
| sup4_ppo_rm3_kl22, sup4_t0.7 | 217 |
| sup4_xl_bo512_rm4, sup4_xl_bo512_rm4 | 217 |
| sup4_ppo_rm3_kl22, sup4_ppo_rm3_kl260 | 216 |
| sup4_ppo_rm3_kl2, sup4_ppo_rm3_kl6 | 215 |
| sup4_bo64_rm3, sup4_bo64_rm3 | 212 |
| ref, sup4_xl_bo1024_rm3 | 212 |
| sup4_xl_bo512_rouge, sup4_xl_bo512_rouge | 204 |
| sup4_bo512_rm3, sup4_bo512_rm3 | 203 |
| sup4_ppo_rm3_kl260, sup4_ppo_rm3_kl69 | 196 |
| sup4_ppo_rm3_kl2, sup4_ppo_rm3_kl260 | 186 |
| sup4_ppo_rm3_kl6, sup4_t0.7 | 184 |
| sup4_ppo_rm3_kl69, sup4_t0.7 | 183 |
| sup4_ppo_rm3_kl260, sup4_t0.7 | 179 |
| ref, sup3 | 158 |
| ref, sup3_bo63_rm2 | 157 |
| sup4_6b_ppo_rm4_6b, sup4_ppo_rm4 | 156 |
| sup3, sup3_bo8_rm2 | 155 |
| sup4_ppo_rm3, sup4_t0.7 | 152 |
| sup3, sup3_ppo_rm1 | 151 |
| sup3_bo63_rm2, sup3_ppo_rm1 | 151 |
| sup3_bo63_rm2, sup3_bo8_rm2 | 148 |
| ref, sup4_6b_t.3 | 146 |
| ref, sup4_6b_t.5 | 144 |
| sup4_ppo_rm4, title | 143 |
| ref, sup4_6b_t1 | 141 |
| sup4_6b_t.3, sup4_6b_t1 | 140 |
| sup4_12b, sup4_ppo_rm4 | 137 |
| sup4_6b_t.5, sup4_6b_t1 | 137 |
| sup4_6b_t.3, sup4_6b_t.5 | 136 |
| ref, sup4_6b_ppo_rm4_6b_t.7 | 135 |
| sup4, sup4_ppo_rm4 | 134 |
| ref, sup4_6b_ppo_rm4_6b_t.5 | 134 |
| sup4_6b_ppo_rm4_6b_t.5, sup4_6b_ppo_rm4_6b_t.7 | 132 |
| sup4_3b, sup4_ppo_rm4 | 132 |
| sup4_6b_ppo_rm4_6b, sup4_6b_ppo_rm4_6b_t.7 | 132 |
| sup4_ppo_rm3, sup4_ppo_rm3 | 131 |
| sup4_12b, sup4_6b | 131 |
| sup4_6b_ppo_rm4_6b, sup4_6b_ppo_rm4_6b_t.5 | 127 |
| ref, sup4_bo256_rm3 | 127 |
| sup4_12b, title | 127 |
| sup4_6b, sup4_6b_ppo_rm4_6b | 127 |

Table 24: The unique comparison pairs and their counts in the *validation* split of the preference dataset. (Part 3)

| policies | count |
|---|---|
| ref, sup4_6b_p.7 | 124 |
| ref, sup4_6b_p.9 | 124 |
| ref, sup4_6b_p.8 | 124 |
| ref, sup4_xl_bo8_rm4 | 124 |
| sup4_bo128_rm3, sup4_bo128_rm3 | 119 |
| sup4_xl_bo16_rm4, sup4_xl_bo8_rm4 | 118 |
| sup4, sup4_12b | 116 |
| pretrain_12b, sup4_6b_ppo_rm4_6b | 114 |
| sup4_6b_p.7, sup4_6b_p.8 | 114 |
| sup4, title | 113 |
| sup4_3b, sup4_6b_ppo_rm4_6b | 113 |
| ref, sup4_xl_bo8_rouge | 113 |
| sup4_6b_p.8, sup4_6b_p.9 | 113 |
| sup4_6b_p.7, sup4_6b_p.9 | 112 |
| sup4_bo8_rm3, sup4_xl_bo16_rm3 | 111 |
| sup4_6b, title | 107 |
| sup4_6b, sup4_ppo_rm4 | 106 |
| sup4_3b, title | 106 |
| sup4_t0.7, sup4_xl_bo2_rouge | 106 |
| sup4_t0.7, sup4_xl_bo2_rm3 | 104 |
| sup4_xl_bo16_rouge, sup4_xl_bo8_rouge | 103 |
| sup4, sup4_3b | 102 |
| sup4_12b, sup4_6b_ppo_rm4_6b | 100 |
| pretrain_12b, sup4_ppo_rm4 | 98 |
| pretrain_3b, sup4 | 97 |
| sup4_6b_ppo_rm4_6b, title | 97 |
| sup4_12b, sup4_3b | 95 |
| sup4_t0.7, sup4_xl_bo2_rm4 | 95 |
| sup4_xl_bo16_rm4_6b, sup4_xl_bo8_rm4_6b | 94 |
| pretrain_6b, sup4_ppo_rm4 | 93 |
| pretrain_12b, sup4_12b | 93 |
| ref, sup4_xl_bo8_rm4_6b | 92 |
| pretrain_6b, sup4_12b | 91 |
| pretrain_6b, title | 90 |
| pretrain_3b, pretrain_6b | 90 |
| pretrain_12b, sup4_3b | 90 |
| pretrain_6b, sup4_6b | 88 |
| sup4_ppo_rm3_kl10, sup4_ppo_rm3_kl6 | 86 |
| sup4_ppo_rm3_kl10, sup4_ppo_rm3_kl2 | 86 |
| sup4_t0.7, sup4_xl_bo2_rm4_6b | 84 |
| sup4, sup4_6b | 84 |
| sup4_3b, sup4_6b | 84 |
| pretrain_12b, sup4_6b | 82 |
| pretrain_6b, sup4_6b_ppo_rm4_6b | 81 |
| sup4, sup4_6b_ppo_rm4_6b | 81 |
| pretrain_3b, sup4_ppo_rm4 | 79 |
| pretrain_3b, sup4_3b | 79 |
| sup4_ppo_rm3_kl10, sup4_ppo_rm3_kl22 | 77 |
| pretrain_3b, sup4_6b_ppo_rm4_6b | 75 |
| pretrain_6b, sup4_3b | 74 |

Table 25: The unique comparison pairs and their counts in the *validation* split of the preference dataset. (Part 4)

| policies | count |
|---|---|
| sup4_6b_t0.7, sup4_bo64_rm3 | 74 |
| pretrain_6b, sup4 | 72 |
| pretrain_xl, sup4_6b_ppo_rm4_6b | 68 |
| sup4_ppo_rm3_kl10, sup4_t0.7 | 67 |
| pretrain_3b, sup4_12b | 67 |
| sup4_ppo_rm3_kl10, sup4_ppo_rm3_kl260 | 65 |
| pretrain_xl, sup4_12b | 64 |
| pretrain_12b, sup4 | 63 |
| sup4_bo128_rm3_6b, sup4_bo256_rm3_6b | 63 |
| ref, sup4_bo128_rm3_6b | 62 |
| ref, sup4_xl_bo16_rm4 | 61 |
| pretrain_3b, title | 60 |
| sup4_6b_t0.7, sup4_bo512_rm3 | 60 |
| pretrain_xl, sup4_3b | 60 |
| sup4_ppo_rm3_kl10, sup4_ppo_rm3_kl69 | 59 |
| pretrain_xl, sup4 | 58 |
| sup4_xl_bo8_rm4, sup4_xl_bo8_rm4 | 58 |
| pretrain_xl, sup4_ppo_rm4 | 56 |
| ref, sup4_xl_bo2_rm3 | 56 |
| ref, sup4_xl_bo16_rm3 | 56 |
| sup4_bo8_rm3, sup4_bo8_rm3 | 56 |
| pretrain_xl, title | 56 |
| ref, sup4_xl_bo2_rouge | 55 |
| ref, sup4_xl_bo16_rouge | 55 |
| sup4_xl_bo8_rouge, sup4_xl_bo8_rouge | 55 |
| pretrain_12b, pretrain_xl | 54 |
| pretrain_12b, pretrain_3b | 51 |
| ref, sup4_xl_bo2_rm4 | 48 |
| pretrain_12b, title | 47 |
| pretrain_3b, pretrain_xl | 47 |
| ref, sup4_xl_bo2_rm4_6b | 47 |
| pretrain_12b, pretrain_6b | 47 |
| ref, sup4_xl_bo16_rm4_6b | 47 |
| sup4_xl_bo8_rm4_6b, sup4_xl_bo8_rm4_6b | 46 |
| pretrain_3b, sup4_6b | 45 |
| pretrain_xl, sup4_6b | 45 |
| pretrain_6b, pretrain_xl | 44 |
| sup4_bo128_rm3_6b, sup4_bo128_rm3_6b | 33 |
| ref, sup4_bo256_rm3_6b | 31 |
| human_editor, sup4_6b_t0.7 | 3 |
| human_editor, ref | 2 |

Table 26: The unique comparison pairs and their counts in the *validation* split of the preference dataset. (Part 5)

| policies | count |
|---|---|
| supcnndm3_6b_t.3, supcnndm3_6b_t.3 | 1410 |
| pretrain_6b_t.7, sup4_6b_ppo_rm4_6b_kl14 | 148 |
| ref, sup4_6b_t0.7 | 148 |
| ref, sup4_6b_ppo_rm4_6b_kl14 | 148 |
| pretrain_6b_t.7, ref | 146 |
| sup4_6b_ppo_rm4_6b_kl14, sup4_6b_t0.7 | 144 |
| pretrain_6b_t.7, sup4_6b_t0.7 | 140 |

Table 27: The unique comparison pairs and their counts in the *validation_cnndm* split of the preference dataset. (Part 5)

