# OpenReview forum: "The N+ Implementation Details of RLHF with PPO: A Case Study on TL;DR Summarization"
_colmweb.org/COLM/2024/Conference — COLM_

### Official Review · Reviewer_toWU · 2024-05-01

**Rating:** 7
**Confidence:** 4
**Ethics Flag:** 2

**Summary:**

The paper discusses more than 20 implementation details of RLHF with PPO using the tl;dr summarization task. The authors achieve great performance on the summarization task, and the authors reproduced the scaling behavior by OpenAI by experimenting with models of different sizes. So, the implementation details must be meaningful.

**Ethics Concerns Details:**

Mildly sexual content in Table 1 but fixable

**Questions To Authors:**

Detail 23 seems important but it’d be great if the authors can elaborate what the values are. Are values for each batch? The past k steps?

**Reasons To Accept:**

In general I believe this discussion is a great contribution to the community, as top-performing models’ RLHF procedures are sometimes hidden and not published.

The authors also promise to open-source the code. The appendix is helpful (with implementation details and detailed results and prompts).

Some of the details are great: for example, “including the leading space in completion” (detail 3) is a common bug in implementation and people rarely openly discuss it, so I really appreciate the authors for writing it out.

Another example is detail 7. It is also great because people don’t usually discuss it often.

Detail 23 (eos trick; if there’s no eos in the trajectory then assign it a negative reward) is also something I used before, but I didn’t hear others discussing it. So it’s great to see some confirmation that others are using it too and it works.

**Reasons To Reject:**

(Detail 19) On DPO vs. PPO reward modeling ability: it’s unclear to me whether the comparison is fair.
For DPO reward model, do you mean using the probabilities as an implicit reward model?
In that case, would it be more fair if you not only include your regular reward model validation accuracies, but also PPO-trained LM’s reward modeling accuracy (in other words, similar to DPO, treat the PPO-trained model as an implicit reward model, and compute the validation accuracies)?

Imo the advantage implementation is quite important but there seems to be minimal discussion.

There have been quite a few recent “implementation details of PPO” online. I thought it would be best if the authors highlight the least known and the most important details, and also point out which detail likely doesn’t matter much.

I’m not entirely sure if the example in Table 1 is appropriate so I leave it to other reviewers / the meta-reviewer. I know we are all adults but I also don’t want to run into this genre unprompted, especially in an academic paper (e.g., losing my virginity, leaving me with a boner).

--

Updated score after rebuttal

---

> ### Author Rebuttal · Authors · 2024-05-31
>
> We thank the reviewer for their in-depth review and are glad they find our paper to be an important contribution to the community and furthering open science. We agree that these details are important and deserve to be discussed.
>
> **On DPO vs PPO RM**
>
> Though the title of the DPO paper is “Your Language Model is Secretly a Reward Model”, only by training with DPO’s objective is this true. In particular, DPO has a formula that allows the trained LLM to be used as a reward model in conjunction with the reference LM), as defined in page 5 of [https://arxiv.org/pdf/2305.18290](https://arxiv.org/pdf/2305.18290):
>
> $$\hat{r}_\theta(x, y) = \beta \log\frac{\pi_\theta(y|x)}{\pi_{\text{ref}}(y|x)}$$
>
> But this isn’t the case for PPO. PPO’s policy LM is trained exclusively to maximize RM and so its logprobs will not correspond to an implicit RM since it isn’t part of the objective function. Since both methods use an RM (DPO’s implicit RM, and PPO’s separate, explicit RM) we compare those.
>
> **Novelty of our details**
>
> The most similar work to ours is Delve into PPO by Zheng et al, 2023. Notably, our work distinguishes itself by building a scaling behavior – we showed our RLHF pipeline scales with model sizes and works across random seeds. Our investigation is also more extensive, and our findings differ (we find reward whitening to be optional and advantage whitening to be beneficial, whereas Zheng et al find the opposite). Though we find the EOS trick to be the most underreported detail, our work proposes a collection of different details to form an overall “correct” setup for the popular benchmark summarization task, which we hope will be adopted by the community. In contrast, Zheng et al focus on the best combination of PPO details for a single model on HH-RLHF.
>
> **Table 1**
>
> This is a good point. We have already changed it to another, more appropriate sample.
>
> ```
> SUBREDDIT: r/relationships
> TITLE: Me [19 F] should I be trying to help my
> brother[16 M] with his life?
> POST: This is my first Reddit post and I’m not
> sure if I’m doing it right
> ```
>
>
>
> **Detail 23 Values**
>
> Are you referring to “values'' in Detail 13 (not 23)? We propose to treat the reward function as a per-token value function and get its outputs at non-eos tokens. We find that, in general, the reward model does not learn to be a per-token RM and gives nonsensical outputs at non-eos tokens.

---

> > ### Comment · Reviewer_toWU · 2024-06-05
> >
> > Thank you for the reply. In the meantime I'm reading through other reviews and I'll update my review accordingly.

---

### Official Review · Reviewer_UfCo · 2024-05-11

**Rating:** 6
**Confidence:** 5
**Ethics Flag:** 1

**Summary:**

This paper summarizes the reproduction details of the OpenAI work: RLHF for summarization. The authors study many different items that are specific and important for reproduction. The authors have found several key points that affect the performance.

**Reasons To Accept:**

1. This is a good manuscript for others to reproduce the work of RLHF for summarization.
2. The tricks are detailed in lots of ways and aspects, which are important for reproduction.
3. The experience could be possible to transfer to other applications.

**Reasons To Reject:**

1. Though the comprehensive study appears, it is somehow hard to show the connections between these different details, this potentially could be crucial.
2. The application summarization task is specific, however, it would be better to make some details that could be transferable. Or it should be study some of these items.
3. The base model GPT-2 is somehow old now, at least LLaMA would be a better replacement.

---

> ### Author Rebuttal · Authors · 2024-05-31
>
> We are glad the reviewer found our work to be a good manuscript to reproduce RLHF summarization.
> > Though the comprehensive study appears, it is somehow hard to show the connections between these different details, this potentially could be crucial.
>
> We address this by releasing the source code and have tried to make it as minimal as possible, reducing the mental burden of understanding how these details connect with each other.
>
> > The application summarization task is specific, however, it would be better to make some details that could be transferable. Or it should be study some of these items.
>
> This is a great point. The main objective of this work is to dive really deep into TL;DR summarization to make sure the RLHF pipeline works (and scales with model sizes). Given this knowledge, our next steps include adding other benchmark tasks such as anthropic’s HH.
>
> > The base model GPT-2 is somehow old now, at least LLaMA would be a better replacement.
>
> We actually used pythia models instead of gpt-2 models. Furthermore, we choose pythia models for several reasons:
>
> * **pythia models compare more fairly with base models used in Stiennon et al., (2020)**. In particular, Pythia models were trained with ~207B tokens, and Stiennon et al., (2020) suggested that they trained the base models for “200-300 billion tokens”. In comparison, even LLaMA 1 models are trained with 1000 to 1400 billion tokens (https://arxiv.org/pdf/2302.13971).
> * **LLaMA models don’t have 1B models**. LLaMA 1 models only have 8B to 60B models, which makes it difficult to conduct the RLHF scaling experiments shown in this paper.
> * **Pythia models are truly open**. Unlike many open weights models like LLaMA, Pythia models release the training code, source code, and are fully transparent.

---

> > ### Comment · Reviewer_UfCo · 2024-06-03
> > **Thanks for the rebuttal**
> >
> > Thanks the authors for the rebuttal. I look forward to your next versions of these details. It's okay for the model side.
> > Therefore, I will raise the score.

---

### Official Review · Reviewer_5BXq · 2024-05-13

**Rating:** 6
**Confidence:** 4
**Ethics Flag:** 1

**Summary:**

This paper contributes a detailed reproduction of OpenAI’s RLHF work in the TL;DR summarization. The nice thing about the work is that many technical details (datasets with tokenizer, truncation, padding, EOS tokenizing, disabling dropout, the way they extract reward from the EOS token, etc) are transparent. This should make the paper a good reference for future effort to replicate RLHF work in the dataset.

The work also demonstrated the scaling behavior of PPO across different model sizes, this is unsurprising though.

**Reasons To Accept:**

* This paper will be a good reference for future work on replicating RLHF work in the TL;DR summarization.

**Reasons To Reject:**

* The novelty of the work is limited. I consider it as a "white" paper that to unblock everyone who tries to replicate "RLHF" work, which is to me a significant contribution. But other than that I don't see other novelty of the work.

---

> ### Author Rebuttal · Authors · 2024-05-31
>
> We are glad the reviewer found our work to be a good reference and appreciates how significant our contribution is. We agree our work does not propose a novel method, but we do provide novel, open results and novel findings about the most common benchmark and baseline in the field.
>
> Many of our details are not commonplace in standard RLHF libraries which highlights their novelty and necessity to researchers e.g. TRL’s PPO 1) does not use reward normalization, 2) does not use a separate value network, 3) does not use the EOS trick, 4) reported issues with negative KL divergence (https://github.com/huggingface/trl/issues/355).
>
> The strength of our open and reproducible baselines is also a novelty. For example, the DPO paper (https://arxiv.org/pdf/2305.18290) reported its PPO model to have 57% max win rate against reference summaries (judged by GPT4). Whereas our same-sized model has a ~80% win rate (in a separate GPT4 eval; just one random seed). By releasing code and baselines, we hope to ground the field in reproducible and significant results.
>
> Finally, we believe there is some novelty in the correctness of our preprocessing and eval guidelines that enables the validity of the benchmark. For example, we are the first to openly replicate the scaling behaviors on summarization and we hope there is novelty for the community to now use this as a benchmark for scaling properties of RLHF algorithms.

---

> ### Author Response · Authors · 2024-06-05
> **Rebuttal Reminder**
>
> Given that the discussion period ends tomorrow, let us know if there are any concerns you feel we haven't addressed or clarifications to improve our score. Thank you!

---

### Official Review · Reviewer_EUPs · 2024-05-23

**Rating:** 6
**Confidence:** 5
**Ethics Flag:** 1

**Summary:**

The paper does a comprehensive reproducibility experiment on the RLHF results from a past OpenAI paper. The authors experiment with different details of training that are often overlooked or not discussed in detail in most papers.

**Reasons To Accept:**

I think this paper is a great read and a guide to all practitioners in the community working in the intersection of LLM and RLHF. It would

**Reasons To Reject:**

I am not sure how much it might align with the "research novelty" aspect of the conference. I leave that judgement to the editors.

I would like to point out though that the authors mentioned that many of the practices are not standard in the community. In that note, it would be good to contrast the design choices of the authors against the out-of-the-box choices presented in many open source libraries like deepspeed-chat, trl, etc. Since the main objective of this paper is to make the engineering practices more reproducible and "open", it is important to contrast the presented choices against other available alternatives.

---

> ### Author Rebuttal · Authors · 2024-05-31
>
> We are glad the reviewer found our work to be a great guide to all practitioners in the community. While we agree our work does not propose a novel method, we believe there is novelty in our investigation of details and the strength of our open and reproducible baselines. Our work helps the community build upon a better foundation (and create better science).
>
> For example, our PPO models are well-calibrated and have a very high GPT4 win rate. The DPO paper (https://arxiv.org/pdf/2305.18290) reported its PPO model to have 57% max win rate against reference summaries (judged by GPT4). Whereas our same-sized model has a ~80% win rate. Our paper can become a strong baseline for future works and reviews to refer to. Below, we take the example completions at Table 5 from the DPO paper and compare them to ours :
>
> **DPO TL;DR:** About to propose to my girlfriend of almost 2 years, but I am extremely
> anxious and can’t sleep. How can I relax?
>
> **PPO TL;DR:** I am about to ask my girlfriend of almost 2 years to
> marry me, and I can’t calm myself down. Any tips on how to relax? I would like to be able to sleep tonight (and the night before)..............................................................................................................................
>
> **Our PPO TL;DR:** I am about to propose to my girlfriend of almost 2 years in a couple days and I am freaking out. I can't seem to relax leading up to it. Any suggestions on how to calm myself down and relax?
>
> As demonstrated, the PPO baseline used in the DPO paper seems broken (with excessive periods), whereas our completion is coherent and does nothing like that.
>
> Comparing deepspeed-chat and trl: This is a good point. We noticed that TRL’s PPO 1) does not use reward normalization, 2) does not use a separate value network, 3) does not use the EOS trick, 4) reported issues with negative KL divergence (https://github.com/huggingface/trl/issues/355). We noticed deepspeed-chat does not use reward normalization.
>
> Overall, direct comparisons are difficult to make because the full-stack setups are very different. It may be better to compare the final model checkpoints, but deepspeed-chat / trl did not release checkpoints for TL;DR summarization (the exact challenge this work aims to address).

---

> ### Author Response · Authors · 2024-06-05
> **Rebuttal Reminder**
>
> Given that the discussion period ends tomorrow, let us know if there are any concerns you feel we haven't addressed or clarifications to improve our score. Thank you!

---

### Official Review · Reviewer_BkcZ · 2024-05-24

**Rating:** 7
**Confidence:** 3
**Ethics Flag:** 1

**Summary:**

The paper proposes a re-implementation of the RLHF algorithm with PPO. The paper provides insights into the key implementation details for the reproduction of OpenAI's seminal TL;DR summarization work. Each key implementation insight has been backed by an ablation study into its effect on the final performance. This thorough examination enhances our understanding of the algorithm's intricacies and will serve as a valuable resource for the research community.

**Questions To Authors:**

Please check my questions above.

**Reasons To Accept:**

The paper's strength lies in its careful study of the different implementation details in RLHF.  The paper demonstrates the scalability of RLHF with various model sizes and validates the effectiveness of the RLHF approach for the pythia models at different scales.

Furthermore, the authors have made the complete source code, model checkpoints, and training metrics publicly available. I believe, future works can build on these models for a deeper dive into the model's behavior with RLHF.

Furthermore, the paper conducts consistency and calibration studies on the trained reward models. The authors showcase the consistency of these models with human preferences, as judged by external metrics like GPT-3.5, and also report comparisons of their trained reward model to dpo's reward model. They point to the specific differences between the two algorithms.

These strengths highlight the paper's contributions to advancing the understanding and application of RLHF, particularly in the context of summarization tasks.

**Reasons To Reject:**

I don't have many concerns with the paper as-is. Here are a few questions about the experiments to the authors.

a) (Detail 2) How do the experiment observations change with different restrictions to the length of the query tokens (say 256 v/s 512 v/s 768)?

b) (Detail 4) How do the differences in SFT and Preference datasets affect the final performance of the trained model? Is there a control study, where the authors restrict the length of the examples in  both the datasets to check their impact on the final performance?

c) (Detail 21) What's the effect of multiple epoch training on the same SFT prompts? Do the authors observe differences in the quality of the different SFT prompts?

d) "Reward whitening makes the model generate shorter outputs." Is this helpful only for summarization tasks?

---

> ### Author Rebuttal · Authors · 2024-05-30
>
> We are glad that the reviewers found our paper contributing to advancing the understanding and application of RLHF. We hope our careful study is helpful for future practitioners when building out their RLHF pipeline.
>
> > a) (Detail 2) How do the experiment observations change with different restrictions to the length of the query tokens (say 256 v/s 512 v/s 768)?
>
> That is a great question! However, it might be difficult to evaluate because the reference summary has less than 53 tokens using the Pythia tokenizer, so comparing with 256 / 512/ 768 tokens might not be as meaningful.
>
> That said, length is definitely a confounding factor, so it motivated us to do length-controlled comparisons in Figures 12 and 13.
>
> > How do the differences in SFT and Preference datasets affect the final performance of the trained model? Is there a control study, where the authors restrict the length of the examples in both datasets to check their impact on the final performance?
>
> A reasonable comparison might be to filter out the portion of the preference dataset that has more than 53 tokens. This might be an interesting experiment for the camera-ready version if our computes allow.
>
> > What's the effect of multiple epoch training on the same SFT prompts? Do the authors observe differences in the quality of the different SFT prompts?
>
> Great question. Multiple epochs on the SFT dataset might cause overfitting (e.g., https://www.fast.ai/posts/2023-09-04-learning-jumps/) In this work, we have used a single epoch SFT training to follow the setting in Stiennon et al., (2020) closely (e.g., “For supervised baselines, … We use a batch size of 128, and run for a single epoch.”). We find preliminary results at 1.3B scale show overfitting at more than one epoch. Nevertheless, performing multiple epochs training would be a good experiment to be included in the camera-ready version
>
> > d) "Reward whitening makes the model generate shorter outputs." Is this helpful only for summarization tasks?
>
> Human preference datasets generally have longer answers as preferred causing length bias, so reward whitening may be a possible tool that can remedy this in regular chat tasks as well.

---

> > ### Comment · Reviewer_BkcZ · 2024-06-05
> >
> > I thank the reviewers for their detailed response to my questions. Looking over the reviews of my fellow reviewers, I see no major concern, other than whether the paper is novel enough. I strongly believe that we need such papers to dive deeper into the design choices of existing algorithms for better understanding of these models. As such, I keep my score as-is but am in favor of acceptance into the conference proceedings.

---

### Comment · Area_Chair_toex · 2024-06-03
**[Area Chair Comment] Author response**

Dear reviewers,

Please take a look at the author's rebuttals and the other reviews for this paper! If the rebuttals addressed your concerns, please let the authors know about this and update your review. If not, please continue to engage with the authors and the other reviewers in the discussion forum.

Please feel free to comment on reviews (or their author responses) other than your own!

Thanks!

---

### Decision · Program_Chairs · 2024-07-10

**Decision:**

Accept

**Comment:**

The paper outlines a re-implementation of the RLHF pipeline from Stiennon et al. 2020. The main contributions of the paper are (1) reproducing the RLHF pipeline and the scaling behaviors in Stiennon et al 2020, (2) a number of implementation details and insights (e.g. benefits of dropping dropout) that are needed for a stable PPO implementation.

Overall, the reviewers agree that this paper contains useful insights for researchers, especially since PPO training is known to be unstable.

Cons:
It is unclear what advantages this provides over existing frameworks like TRL. The paper is also missing a through comparison with the design choices between these frameworks.